# Long-Tailed Recognition via Information-Preservable Two-Stage Learning

**Fudong Lin & Xu Yuan**[*]
Department of Computer & Information Sciences
University of Delaware
Newark, DE 19711

## Abstract

The imbalance (or long-tail) is the nature of many real-world data distributions, which often induces the undesirable bias of deep classification models toward frequent classes, resulting in poor performance for tail classes. In this paper, we propose a novel two-stage learning approach to mitigate such a majority-biased tendency while preserving valuable information within datasets. Specifically, the first stage proposes a new representation learning technique from the information theory perspective. This approach is theoretically equivalent to minimizing intra-class distance, yielding an effective and well-separated feature space. The second stage develops a novel sampling strategy that selects mathematically informative instances, able to rectify majority-biased decision boundaries without compromising a model's overall performance. As a result, our approach achieves state-of-the-art performance across various long-tailed benchmark datasets. Our code is available at https://github.com/fudong03/BNS_IPDPP.

## 1 Introduction

Class imbalance naturally arises in real-world scenarios, spanning across such wide applications as online transactions, medical diagnoses, social networks spam detection, among many others. Such data in real scenarios usually follows a long-tailed distribution, *i.e.*, a few head classes dominate the entire dataset while some tail classes account for only a small portion. When encountered with such long-tailed data, deep neural networks (DNNs) suffer from majority-biased decision boundaries [31, 53, 24, 57, 13, 41, 16, 42], undesirably favoring frequent classes and leading to poor performance in tail classes. Misclassifying tail classes may yield catastrophic consequences, *e.g.*, failing to identify lung cancer from millions of biomedical images may result in fatalities. To this end, building functional classifiers with unbiased decision boundaries when tackling the long-tailed data is critical but remains open.

So far, the mainstream strategies addressing long-tailed recognition primarily fall into two categories: one-stage learning and two-stage learning methods. One-stage learning approaches include: i) re-weighting strategies [38, 10, 3, 9, 47], which prevent dominant head classes from overwhelming the training process by reducing the weight of loss functions for majority samples; and ii) sampling techniques [4, 21, 58, 25, 15], which create balanced training subsets either by downsampling majority samples or by synthesizing minority samples. However, such strategies struggle to achieve effective performance across both head and tail classes due to their limited representation learning capabilities. On the other hand, two-stage learning methods [28, 51, 11, 62, 30, 39, 34, 35, 27] decouple the training process into "representation learning" and "classification" stages. The former stage focuses on learning effective and generalizable feature spaces, while the latter stage aims to rectify majority-biased decision boundaries caused by highly skewed data distributions.

---

[*]Corresponding author: Dr. Xu Yuan (xyuan@udel.edu)

39th Conference on Neural Information Processing Systems (NeurIPS 2025).

However, previous two-stage learning approaches fail to deliver satisfactory performance in long-tailed recognition, primarily due to the following two limitations. First, in the first stage, conventional representation learning methods [5, 23, 1, 22, 36] face significant challenges in handling long-tailed data, resulting in poor feature representations for both head and tail classes. Recent advancements targeting long-tailed recognition [39, 34, 35, 27] have mitigated these limitations to some extent, able to produce higher-quality feature spaces. Unfortunately, they struggle to learn effective feature spaces for securing clear separation between head and tail classes, ultimately limiting their overall performance. Besides, the second stage is hindered by the absence of effective sampling techniques. Oversampling methods [4, 49, 58, 25, 15] often suffer from mode collapse, producing synthetic minority samples with limited diversity. Conversely, undersampling approaches [21, 40, 61] frequently lead to significant information loss due to the removal of a substantial portion of majority samples, severely compromising the model's overall performance.

In this work, we advance two-stage learning through two key contributions: i) a novel representation learning strategy, namely **Balanced Negative Sampling (BNS)**, able to learn effective and well-separated feature spaces; and ii) a new sampling technique called **Information-Preservable Determinantal Point Process (IP-DPP)**, able to balance the training subsets with informative instances, for mitigating the majority-biased tendency. Our key idea involves first training an effective feature extractor using our BNS method. This feature extractor is then fine-tuned on balanced subsets sampled through our IP-DPP approach to handle imbalanced classification. Specifically, our BNS approach constructs an effective and well-separated representation space by maximizing mutual information between two augmented views of the original data. Formal proof is provided showing our solution is theoretically equivalent to minimizing intra-class distance. This allows the model to capture instance-level semantics, ensuring high-quality feature representations, while also learning class-level semantics to achieve well-separated feature spaces. Besides, our IP-DPP method, inspired by Shannon information theory, is designed to sample balanced subsets that prioritize mathematically informative instances. As a result, it excels in rectifying majority-biased decision boundaries while maintaining the model's overall performance.

## 2 Related Work

This work adopts a two-stage learning paradigm, in which Balanced Negative Sampling (BNS) is designed to learn an effective representation space, while the Information-Preservable Determinantal Point Process (IP-DPP) is introduced to select mathematically informative samples. We next discuss how our approaches relate to, and differ from, prior solutions.

**Representation learning methods** are typically employed in the first stage to construct high-quality feature spaces. Previous studies include KCL [34], which devises $k$-positive contrastive learning for learning balanced feature spaces, TSC [35], which improves the uniformity of feature spaces via targeted supervised contrastive learning, SBCL [27], which proposes subclass-balancing contrastive learning for instance- and subclass-balanced feature spaces, among many others [5, 6, 23, 7, 1, 22, 36, 37]. However, conventional representation learning methods result in suboptimal representations for both head and tail classes, while those specifically designed for long-tailed recognition struggle to achieve well-separated feature spaces. Differently, our BNS method frames representation learning from the information theory perspective. This solution is mathematically equivalent to minimizing intra-class distance, thereby resulting in effective and well-separated feature spaces.

**Sampling methods** are commonly used in the second stage to rectify biased decision boundaries. Existing techniques can be roughly categorized into oversampling and undersampling approaches. Oversampling methods balance class priors by generating minority samples. Traditional methods [4, 19, 20, 54] apply linear combinations of existing instances to synthesize the minority data, but they fail to respect the non-linear structure of synthetic data. Recent solutions [49, 17, 58, 50, 25, 15] leverage deep generative models to capture the non-linear structures in data synthesis. However, these methods suffer from mode collapse, resulting in synthesized minority samples with limited diversity. Undersampling approaches [21, 40, 61], by contrast, create balanced subsets by removing a substantial portion of majority samples. However, this process causes significant information loss, severely impacting the model's overall performance. Our IP-DPP method falls into this category. It addresses information loss by sampling mathematically informative instances, thereby effectively rectifying biased decision boundaries while maintaining the model's overall performance.

# 3 Preliminary

**Mutual Information (MI)** refers to a measure of the mutual dependence between two random variables. It quantifies the amount of information that knowing one random variable reduces the uncertainty of the other variable. Given two jointly discrete random variables $\mathbb{X}$ and $\mathbb{Y}$, for their mutual information, we have:

$$MI(\mathbb{X}, \mathbb{Y}) = \sum_{\boldsymbol{x} \in \mathbb{X}} \sum_{\boldsymbol{y} \in \mathbb{Y}} p_{\mathbb{X}\mathbb{Y}}(\boldsymbol{x}, \boldsymbol{y}) \log \frac{p_{\mathbb{X}\mathbb{Y}}(\boldsymbol{x}, \boldsymbol{y})}{p_{\mathbb{X}}(\boldsymbol{x}) p_{\mathbb{Y}}(\boldsymbol{y})}. \tag{1}$$

Here, $p_{\mathbb{X}\mathbb{Y}}(\boldsymbol{x}, \boldsymbol{y})$ represents the joint probability of $\mathbb{X}$ and $\mathbb{Y}$, while $p_{\mathbb{X}}(\boldsymbol{x})$ and $p_{\mathbb{Y}}(\boldsymbol{y})$ are the marginal probabilities of $\mathbb{X}$ and $\mathbb{Y}$, respectively.

**Information Content (IC)**, also known as Shannon information or self-information, measures the degree of "surprise" associated with a particular outcome. Given a ground set $\mathbb{X}$, for any event $\boldsymbol{x} \in \mathbb{X}$ with probability $p(\boldsymbol{x})$, the information content $I(\cdot)$ is defined as follows:

$$I(\boldsymbol{x}) = -\log\left[p(\boldsymbol{x})\right]. \tag{2}$$

By definition, information content has three key properties: i) an event with $100\%$ probability yields no information; ii) less probable events are more surprising and contain more information; and iii) given a set of independent events, its total self-information equals the sum of each event's individual self-information, *i.e.*, $I(\mathbb{X}) = \sum_{\boldsymbol{x} \in \mathbb{X}} I(\boldsymbol{x})$.

# 4 Our Approaches

## 4.1 Problem Statement

Consider a set of $N$ samples for training, *i.e.*, $\mathbb{X} = \{(\boldsymbol{x}_i, \ y_i)\}_{i=1}^{N}$, where data point $\boldsymbol{x}_i$ is labeled with $y_i$. Suppose the training set has $C$ classes, *i.e.*, $y \in \{1, 2, \ldots, C\}$, and let $N_c \ (c = 1, 2, \cdots, C)$ be the number of training data for the $c$-th class. Without loss of generality, we consider all classes to be sorted in the decreasing order, *i.e.*, $N_c \geq N_{c+1}$. Naturally, $\forall N_c$, we have $N_c \geq N_C$. Here, we consider a long-tailed setting, *i.e.*, $N_1 \gg N_C$, indicating that head and tail classes are highly skewed.

Deep neural networks (DNNs) struggle with such long-tailed data, where they perform poorly on tail classes. This can be attributed to two primary factors. First, when using conventional representation learning methods, the inherent "label bias" in long-tailed data leads to poor feature spaces. Second, majority class samples tend to dominate the training process, resulting in biased decision boundaries that unfairly favor the head classes. These two factors call for the development of innovative solutions in both representation learning and classification stages.

## 4.2 Stage 1: Representation Learning via Balanced Negative Sampling

Our first stage aims to learn an effective and well-separated feature space for long-tailed recognition. We resort to maximizing mutual information between instances sharing the same label—a process mathematically equivalent to minimizing intra-class distances.

**Our Objective.** Given any image of the ground set $\boldsymbol{x} \in \mathbb{X}$, we employ data augmentation modules to transform it into two different views of the same sample, denoted as $\boldsymbol{x}_i \in \mathbb{X}_Q$ and $\boldsymbol{x}_j \in \mathbb{X}_V$. Let $f_{\boldsymbol{\theta}}(\cdot)$ be a DNN parameterized by $\boldsymbol{\theta}$, used for encoding feature representations. Here, we regard $\boldsymbol{Q}(\mathbb{X}_Q; \boldsymbol{\theta})$ and $\boldsymbol{V}(\mathbb{X}_V; \boldsymbol{\theta})$ as feature spaces corresponding to $\mathbb{X}_Q$ and $\mathbb{X}_V$, respectively. In this work, our goal is to learn high-quality feature representations for imbalanced classification by maximizing the mutual information between two feature spaces:

$$\underset{\boldsymbol{\theta}}{\arg\max} \ MI(\boldsymbol{Q}(\mathbb{X}_Q; \boldsymbol{\theta}), \boldsymbol{V}(\mathbb{X}_V; \boldsymbol{\theta})). \tag{3}$$

In the rest of the paper, we abbreviate $\boldsymbol{Q}(\mathbb{X}_Q; \boldsymbol{\theta})$ and $\boldsymbol{V}(\mathbb{X}_V; \boldsymbol{\theta})$ as $\boldsymbol{Q}$ and $\boldsymbol{V}$, respectively. Then, given any images of $\boldsymbol{x}_i \in \mathbb{X}_Q$ and $\boldsymbol{x}_j \in \mathbb{X}_V$, their representations can be expressed respectively as $\boldsymbol{q}_i \in \boldsymbol{Q}$ and $\boldsymbol{v}_j \in \boldsymbol{V}$, where $\boldsymbol{q}_i = f_{\boldsymbol{\theta}}(\boldsymbol{x}_i)$ and $\boldsymbol{v}_j = f_{\boldsymbol{\theta}}(\boldsymbol{x}_j)$.

**CL-based Representation Learning.** However, directly maximizing the mutual information between two representation spaces $\boldsymbol{Q}$ and $\boldsymbol{V}$ is computationally intractable. Worse still, prior studies [60, 39]

have demonstrated that long-tailed data inherently causes "label bias", resulting in poor representations for tail classes. In this work, we reformulate Eq. (3) within the framework of contrastive learning (CL), enabling the efficient optimization of our objective. Meanwhile, employing CL-based techniques [5, 23, 29, 37, 27] can also effectively mitigate the "label bias" issue, as highlighted in prior studies [60, 39].

Specifically, our approach leverages a binary classifier to distinguish the target image from noise samples, drawing inspiration from Noise Contrastive Estimation (NCE) [18]. Given an anchor image $\boldsymbol{x}_i \in \mathbb{X}_Q$, we have its corresponding target image $\boldsymbol{x}_{j,i}^+ \in \mathbb{X}_V$. Here, $\boldsymbol{x}_i$ and $\boldsymbol{x}_{j,i}^+$ are two augmented versions of the same image. Meanwhile, we sample $n$ noise images (having different labels with $\boldsymbol{x}_i$) from the dataset $\mathbb{X}_V$, denoted as $\{\boldsymbol{x}_j^-\}_{j=1}^n$. Regarding their representations, we have one positive pair $(\boldsymbol{q}_i, \boldsymbol{v}_{j,i}^+)$ and $n$ negative pairs $\{(\boldsymbol{q}_i, \boldsymbol{v}_j^-)\}_{j=1}^n$. Therefore, there is a $\frac{1}{n+1}$ chance to pick the positive pair and a $\frac{n}{n+1}$ chance to pick the negative pair. Let $p(\cdot)$ be the joint probability of $\boldsymbol{Q}$ and $\boldsymbol{V}$, and $g(\cdot)$ represent a binary classifier, where its output $d = 1$ and $d = 0$ denote the positive and negative pair, respectively. Then, we have:

$$g(\boldsymbol{q}_i, \boldsymbol{v}_j \mid d) = \begin{cases} \frac{1}{n+1} \, p(\boldsymbol{q}_i, \boldsymbol{v}_{j,i}^+), & d = 1 \\ \frac{n}{n+1} \, p(\boldsymbol{q}_i, \boldsymbol{v}_j^-), & d = 0 \end{cases}. \tag{4}$$

Considering the positive pair only, we arrive at:

$$g(\boldsymbol{q}_i, \boldsymbol{v}_j \mid d = 1) = \frac{p(\boldsymbol{q}_i, \boldsymbol{v}_{j,i}^+)}{p(\boldsymbol{q}_i, \boldsymbol{v}_{j,i}^+) + n \times p(\boldsymbol{q}_i, \boldsymbol{v}_j^-)}. \tag{5}$$

We assume that the distributions between different classes are independent. Then, for any negative pair $(\boldsymbol{q}_i, \boldsymbol{v}_j^-)$, we have $p(\boldsymbol{q}_i, \boldsymbol{v}_j^-) = p(\boldsymbol{q}_i)p(\boldsymbol{v}_j^-)$. As such, we can rewrite Eq. (5) as follows:

$$g(\boldsymbol{q}_i, \boldsymbol{v}_j \mid d = 1) = \frac{p(\boldsymbol{q}_i, \boldsymbol{v}_{j,i}^+)}{p(\boldsymbol{q}_i, \boldsymbol{v}_{j,i}^+) + n \times p(\boldsymbol{q}_i)p(\boldsymbol{v}_j^-)}. \tag{6}$$

Taking the logarithm of Eq. (6) and rearranging the terms (see Appendix A.1 for detailed derivation), we obtain:

$$\log g(\boldsymbol{q}_i, \boldsymbol{v}_j \mid d = 1) \leq \log \frac{p(\boldsymbol{q}_i, \boldsymbol{v}_{j,i}^+)}{p(\boldsymbol{q}_i)p(\boldsymbol{v}_j^-)} - \log n. \tag{7}$$

Taking the expectation of $p(\boldsymbol{q}_i, \boldsymbol{v}_{j,i}^+)$ on both sides, we have:

$$\mathbb{E}_{p(\boldsymbol{q}_i, \boldsymbol{v}_{j,i}^+)} \log \frac{p(\boldsymbol{q}_i, \boldsymbol{v}_{j,i}^+)}{p(\boldsymbol{q}_i)p(\boldsymbol{v}_j^-)} \geq \mathbb{E}_{p(\boldsymbol{q}_i, \boldsymbol{v}_{j,i}^+)} \log g(\boldsymbol{q}_i, \boldsymbol{v}_j \mid d = 1) + \log n. \tag{8}$$

Combining Eq. (1), Eq. (3), and Eq. (8), we have:

$$\underbrace{MI\,(\boldsymbol{Q}, \boldsymbol{V})}_{\text{maximize MI}} \geq \underbrace{\mathbb{E}_{p(\boldsymbol{q}_i, \boldsymbol{v}_{j,i}^+)} \log g(\boldsymbol{q}_i, \boldsymbol{v}_j \mid d = 1)}_{\text{maximize lower bound}} + \log n. \tag{9}$$

Here, $\log n$ is a constant, indicating that maximizing the lower bound in Eq. (9) is equivalent to maximizing the mutual information between the two feature spaces.

**Balanced Negative Sampling (BNS).** Inspired by prior studies [18, 48, 52], we train a logistic regression classifier to maximize the lower bound in Eq. (9). However, $\log g(\boldsymbol{q}_i, \boldsymbol{v}_j \mid d = 1)$ is computationally intractable. To address this issue, we approximate the classifier's output with sigmoid function $\sigma(\cdot)$, expressed as below:

$$g(\boldsymbol{q}_i, \boldsymbol{v}_j \mid d) = \begin{cases} \sigma(\frac{\boldsymbol{q}_i^\top \boldsymbol{v}_j}{\tau}), & d = 1 \\ \sigma(-\frac{\boldsymbol{q}_i^\top \boldsymbol{v}_j}{\tau}), & d = 0 \end{cases}. \tag{10}$$

Here, $\tau$ is the temperature parameter that controls the sharpness of the similarity scores. As such, our NS-based contrastive learning, designed for learning high-quality representations, is formulated as follows:

$$\mathcal{L}_{\text{NS}} = - \left[ \log \sigma(\frac{\boldsymbol{q}_i^\top \boldsymbol{v}_{j,i}^+}{\tau}) + \sum_{j=1}^n \log \sigma(-\frac{\boldsymbol{q}_i^\top \boldsymbol{v}_j^-}{\tau}) \right]. \tag{11}$$

Our NS-based contrastive learning, *i.e.*, Eq. (11), can mitigate "label bias" inherent to long-tailed data, yielding a higher quality of representations. However, it is ineffective in learning a well-separated

representation space. This is because positive pairs of head classes dominate the representation space. We then propose a novel *Balanced Negative Sampling (BNS)* to learn an effective and well-separated representation space. That is, for a given anchor image $\boldsymbol{x}_i \in \mathbb{X}_Q$, we sample an additional set of $m$ images from $\mathbb{X}_Q$ that share the same label as $\boldsymbol{x}_i$, denoted as $\{\boldsymbol{x}_k\}_{k=1}^m$. Let $\boldsymbol{q}_k \in \boldsymbol{Q}_{i,m}^+$ denote representations for the addition set of images. Then, we have $m+1$ positive pairs $\{(\boldsymbol{q}_*, \boldsymbol{v}_{j,i}^+) \mid \boldsymbol{q}_* \in \{\boldsymbol{q}_i\} \cup \boldsymbol{Q}_{i,m}^+\}$ and $n(m+1)$ negative pairs $\{(\boldsymbol{q}_*, \boldsymbol{v}_j^-) \mid \boldsymbol{q}_* \in \{\boldsymbol{q}_i\} \cup \boldsymbol{Q}_{i,m}^+ \text{ and } j = 1, 2, \ldots, n\}$. Mathematically, our BNS technique can be expressed as:

$$\mathcal{L}_{\text{BNS}} = -\frac{1}{m+1}\left[ \sum_{q_* \in \{q_i\} \cup \boldsymbol{Q}_{i,m}^+} \log \sigma(\frac{\boldsymbol{q}_*^\top \boldsymbol{v}_{j,i}^+}{\tau}) + \sum_{q_* \in \{q_i\} \cup \boldsymbol{Q}_{i,m}^+} \sum_{j=1}^n \log \sigma(-\frac{\boldsymbol{q}_*^\top \boldsymbol{v}_j^-}{\tau}) \right]. \quad (12)$$

In practice, $m$ is set to a small value due to the limited number of samples in minority classes. Eq. (12) effectively enhances the quality of feature representations by maximizing the mutual information shown in Eq. (3). This maximization of mutual information directly corresponds to minimizing intra-class distances, as stated next.

**Theorem 4.1.** *(Intra-Class Distance Mutual Information Theorem) Let $\mathbb{X}_Q^c$ and $\mathbb{X}_V^c$ be two sets of images with the same label c, obtained by different data augmentation techniques. Given a feature extractor $f_{\boldsymbol{\theta}}(\cdot)$, we define $\boldsymbol{Q}^c$ and $\boldsymbol{V}^c$ as the representation spaces for $\mathbb{X}_Q^c$ and $\mathbb{X}_V^c$, respectively. Then, any pair of $\boldsymbol{q}_i^c \in \boldsymbol{Q}^c$ and $\boldsymbol{v}_j^c \in \boldsymbol{V}^c$ is a positive pair. Let $MI(\cdot)$ and $D(\cdot)$ respectively denote the mutual information and a distance metric, we have:*

$$\max MI(\boldsymbol{Q}^c, \boldsymbol{V}^c) \propto \min D(\boldsymbol{Q}^c, \boldsymbol{V}^c), \quad (13)$$

*where $D(\boldsymbol{Q}^c, \boldsymbol{V}^c)$ can be considered as the intra-class distance because they have the same label.*

The proof of Theorem 4.1 is deferred to Appendix A.2.

**Instance-Level and Class-Level Semantics.** An effective representation space must capture two key aspects: instance-level semantics to ensure high-quality feature representations and class-level semantics to achieve well-separated feature spaces. To understand how our BNS technique achieves this, we decompose Eq. (12) as follows:

$$\mathcal{L}_{\text{BNS}} = -\frac{1}{m+1}\left\{ \underbrace{\log \sigma(\frac{\boldsymbol{q}_i^\top \boldsymbol{v}_{j,i}^+}{\tau}) + \sum_{j=1}^n \log \sigma(-\frac{\boldsymbol{q}_i^\top \boldsymbol{v}_j^-}{\tau})}_{\text{instance-level}} + \underbrace{\sum_{q_k \in \boldsymbol{Q}_{i,m}^+} \left[ \log \sigma(\frac{\boldsymbol{q}_k^\top \boldsymbol{v}_{j,i}^+}{\tau}) + \sum_{j=1}^n \log \sigma(-\frac{\boldsymbol{q}_k^\top \boldsymbol{v}_j^-}{\tau}) \right]}_{\text{class-level}} \right\}. \quad (14)$$

According to Theorem 4.1, our BNS approach naturally minimizes distances at both the instance and class levels. Specifically, the pair $(\boldsymbol{q}_i, \boldsymbol{v}_{j,i}^+)$ originates from the same instance, while the pairs $(\boldsymbol{q}_k, \boldsymbol{v}_{j,i}^+)$ are from the same class. This dual-level minimization naturally encourages the emergence of both instance-level and class-level semantics within the representation space, leading to improved representation quality and better separation of feature spaces.

### 4.3 Stage 2: Information-Preservable Determinantal Point Process

Next, we propose a new sampling solution, namely *Information-Preservable Determinantal Point Process (IP-DPP)*, aiming to rectify majority-biased classification decision boundaries while maintaining the model's overall performance. Specifically, our approach builds on the Determinantal Point Process (DPP) [33], a stochastic process that captures global negative correlations, as outlined below.

**Definition 4.2** (Determinantal Point Process). Given a ground set $\mathbb{X}$ with $N$ items, a point process $\mathcal{P}$ in this ground set is a distribution over discrete and finite subsets of $\mathbb{X}$. Let $\boldsymbol{K} \in \mathbb{R}^{N \times N}$ be a real, symmetric marginal kernel matrix indexed by the elements of $\mathbb{X}$. A point process $\mathcal{P}$ is called a DPP only if, for every random subset $\mathbb{Y} \subseteq \mathbb{X}$ drawn according to $\mathcal{P}$, we have:

$$\mathcal{P}(\mathbb{Y}) = \det(\boldsymbol{K}_{\mathbb{Y}}), \quad (15)$$

where $\boldsymbol{K}_{\mathbb{Y}} = [\boldsymbol{K}_{ij}]_{i,j \in \mathbb{Y}}$ is the principle submatrix of $\boldsymbol{K}$, indexed by elements of $\mathbb{Y}$.

Since $\mathcal{P}$ is a probability measurement, *i.e.*, $0 \leq \mathcal{P} \leq 1$, the marginal kernel matrix $\boldsymbol{K}$ must satisfy specific structural properties, as stated next.

*Remark* 4.3 (Properties of Marginal Kernel Matrix). The marginal kernel matrix $\boldsymbol{K}$ for a DPP must satisfy: i) $\boldsymbol{K}$ is a positive semidefinite matrix; and ii) All eigenvalues of $\boldsymbol{K}$ are bounded in the interval $[0, 1]$, *i.e.*, $\boldsymbol{0} \preceq \boldsymbol{K} \preceq \boldsymbol{I}$.

However, it is very difficult to construct a DPP through the marginal kernel matrix $\boldsymbol{K}$ in real long-tailed settings. In this work, we follow the prior study [33] by constructing a DPP based on the $L$-ensemble framework [2]. Specifically, consider an image $\boldsymbol{x}_i \in \mathbb{X}$ with ground truth label $y_i$, let $p(i) = p_{\boldsymbol{\phi}}(y_i | \boldsymbol{x}_i)$ denote the probability of correctly predicting $y_i$ given $\boldsymbol{x}_i$, where the classifier is parameterized by $\boldsymbol{\phi}$. For any two distinct elements $i, j \in \mathbb{X}$, let $p(i, j)$ denote the joint probability of correctly classifying both elements. Assuming independence between classifications, we have $p(i, j) = p(i)p(j)$. Let $\boldsymbol{S}$ be a $N \times N$ matrix, where each element $\boldsymbol{S}_{i,j}$ is defined as follows:

$$\boldsymbol{S}_{i,j} = \begin{cases} \frac{p(i)p(j)}{N}, & i \neq j \\ 1 - \sum_{k \neq j} \frac{p(k)p(j)}{N}, & i = j \end{cases}. \tag{16}$$

As such, $\boldsymbol{S}$ is a symmetric stochastic matrix where each row (or column) sums to 1. The symmetric stochastic matrix $\boldsymbol{S}$ is positive semi-definite, and all its eigenvalues are bounded in $[0, 1]$, as stated in Lemmas 4.4 and 4.5, respectively.

**Lemma 4.4.** *(Positive Semi-definiteness) Let $p(i) = p(y_i | \boldsymbol{x}_i)$ represent the probability of $y_i$ given $\boldsymbol{x}_i$. $\boldsymbol{S} \in \mathbb{R}^{N \times N}$ is a symmetric stochastic matrix, where each row (or column) sums to 1. Then, we have:*

$$\boldsymbol{v}^{\top} \boldsymbol{S} \boldsymbol{v} \geq 0, \quad \forall \boldsymbol{v} \in \mathbb{R}^N. \tag{17}$$

*In other words, $\boldsymbol{S}$ is positive semi-definite.*

**Lemma 4.5.** *(Bounds on Eigenvalues) Let $\{\lambda_i\}_{i=1}^N$ be the eigenvalues of the symmetric stochastic matrix $\boldsymbol{S} \in \mathbb{R}^{N \times N}$, we have:*

$$0 \leq \lambda_i \leq 1, \quad \forall \lambda_i. \tag{18}$$

The proofs of Lemmata 4.4 and 4.5 are deferred to Appendix A.3 and Appendix A.4, respectively.

As such, the symmetric stochastic matrix $\boldsymbol{S}$ satisfies the two properties stated in Remark 4.3. Hence, it can be used to construct a DPP through $L$-ensemble, expressed as below:

$$\mathcal{P}_{\boldsymbol{S}}(\mathbb{Y}) = \frac{\det(\boldsymbol{S}_{\mathbb{Y}})}{\det(\boldsymbol{S} + \boldsymbol{I})}, \tag{19}$$

where $\boldsymbol{I}$ is an $N \times N$ identity matrix. Since $\mathcal{P}_{\boldsymbol{S}}(\mathbb{Y})$ is a probability measurement, it needs to be bounded in $[0, 1]$. The DPP defined in Eq. (19) is valid, *i.e.*, $0 \leq \mathcal{P}_{\boldsymbol{S}}(\mathbb{Y}) \leq 1$, as outlined below.

**Theorem 4.6.** *(Bounded Determinant Probability Measurement) Let $\mathbb{X}$ be a ground set with $N$ items and $\boldsymbol{S} \in \mathbb{R}^{N \times N}$ denote a symmetric stochastic matrix, indexed by elements in $\mathbb{X}$. Here, $\boldsymbol{S}$ is positive semi-definite and satisfies $\boldsymbol{0} \preceq \boldsymbol{S} \preceq \boldsymbol{I}$, where $\boldsymbol{I}$ is the $N \times N$ identity matrix. Let $\boldsymbol{S}_{\mathbb{Y}}$ denote the principal submatrix of $\boldsymbol{S}$ corresponding to $\mathbb{Y}$, for any subset $\mathbb{Y} \subseteq \mathbb{X}$, the following holds:*

$$0 \leq \frac{\det(\boldsymbol{S}_{\mathbb{Y}})}{\det(\boldsymbol{S} + \boldsymbol{I})} \leq 1. \tag{20}$$

*In other words, $\mathcal{P}_{\boldsymbol{S}}(\mathbb{Y}) = \frac{\det(\boldsymbol{S}_{\mathbb{Y}})}{\det(\boldsymbol{S}+\boldsymbol{I})}$ defines a valid probability measurement.*

The proof of Theorem 4.6 is deferred to Appendix A.5.

**Information-Preservable Property.** In the long-tailed settings, our DPP method defined in Eq. (19) can effectively preserve valuable information, as discussed next.

*Remark* 4.7 (Information-Preserving Sampling Principle). Let $I(\boldsymbol{x}) = -\log[p(y|\boldsymbol{x})]$ denote the information content of item $\boldsymbol{x}$ relevant to its correct classification. $\mathcal{P}_{\boldsymbol{S}}(\mathbb{Y} \cup \{\boldsymbol{x}\})$ denotes the probability that item $\boldsymbol{x}$ is sampled by our DPP approach, which prioritizes sampling elements with higher information content, as expressed by:

$$\mathcal{P}_{\boldsymbol{S}}\left(\mathbb{Y} \cup \{\boldsymbol{x}\}\right) \propto I(\boldsymbol{x}). \tag{21}$$

Here, we use a simple example to illustrate how Remark 4.7 holds. Let $\boldsymbol{A} = \{i, j\}$ be a subset of $\mathbb{X}$ sampled by our DPP approach. For the given ground set, $\det(\boldsymbol{S} + \boldsymbol{I})$ is a constant. Then, we arrive at (see Appendix A.6 for details):

$$\mathcal{P}_{\boldsymbol{S}}(\boldsymbol{A}) = \frac{\det(\boldsymbol{S}_{\boldsymbol{A}})}{\det(\boldsymbol{S} + \boldsymbol{I})} \propto \det(\boldsymbol{S}_{\boldsymbol{A}}) = 1 - p(i) \cdot p(j). \tag{22}$$

---
**Algorithm 1** IP-DPP
---
1: **Input:** a ground set $\mathbb{X} = \{\boldsymbol{x}_i\}_{i=1}^N$, its symmetric stochastic matrix $\boldsymbol{S}$, and sample size $k$
2: **Initialize:** standard basis vectors $\{\boldsymbol{e}_i\}_{i=1}^N$ and pairs of orthonormal eigenvalues and eigenvectors $\{(\lambda_i, \boldsymbol{v}_i)\}_{i=1}^N$ for $\boldsymbol{S}$
3: $\boldsymbol{V} \leftarrow \emptyset$
4: **for** $i = 1, 2, \cdots, N$ **do**
5:    **if** $u \sim U(0,1) < \frac{\lambda_i}{\lambda_i + 1}$ **then**
6:       $\boldsymbol{V} \leftarrow \boldsymbol{V} \cup \{\boldsymbol{v}_i\}$
7:       $k \leftarrow k - 1$
8:    **end if**
9:    **if** $k = 0$ **then**
10:       **break**
11:    **end if**
12: **end for**
13: $\mathbb{Y} \leftarrow \emptyset$
14: **while** $|\boldsymbol{V}| > 0$ **do**
15:    **for** $i = 1, 2, \cdots, N$ **do**
16:       $p(i) \leftarrow \frac{1}{|\boldsymbol{V}|} \sum_{\boldsymbol{v} \in \boldsymbol{V}} (\boldsymbol{v}^\top \boldsymbol{e}_i)^2$
17:    **end for**
18:    $i^* \leftarrow \arg\max_i p(i)$
19:    $\mathbb{Y} \leftarrow \mathbb{Y} \cup \{\boldsymbol{x}_{i^*}\}$
20:    $\boldsymbol{V} \leftarrow \boldsymbol{V}_\perp$ // Update $\boldsymbol{V}$ to an orthonormal basis for the subspace orthogonal to $\boldsymbol{e}_{i^*}$
21: **end while**
22: **Return:** a subset $\mathbb{Y}$
---

Therefore, we obtain $\mathcal{P}_{\boldsymbol{S}}(\{i,j\}) \propto -p(i) \cdot p(j)$. In this work, we have $p(i) = p(y_i|\boldsymbol{x}_i)$, implying that images less likely to be correctly classified are more likely to be sampled by our DPP approach. According to information content (see Eq. (2) for details), we have $I(\boldsymbol{x}_i) = -\log[p(y_i|\boldsymbol{x}_i)]$. Thus, $\mathcal{P}_{\boldsymbol{S}}(\{\boldsymbol{x}_i\}) \propto I(\boldsymbol{x}_i)$.

**Balanced Sample Size.** To effectively rectify biased decision boundaries, the cardinality of sampled subsets must be carefully balanced. A subset with a large sample size risks preserving the original imbalance, whereas an overly small subset may lead to significant information loss. Given a ground set with $N$ items, we theoretically demonstrate that that the expected sample size of a DPP defined in Eq. (19) is $N(1 - \ln 2)$. Due to the page limit, the details of this theorem are deferred to Appendix A.7.

This reduction to roughly one-third of the original size is inadequate for balancing the class priors in a highly imbalanced setting. To address this issue, we propose *Information-Preservable Determinantal Point Process (IP-DPP)* to sample balanced subsets by selecting a fixed cardinality $k$ instances from each majority class, as defined below:

$$\mathcal{P}_{\boldsymbol{S}}^k(\mathbb{Y}) = \frac{\det(\boldsymbol{S}_{\mathbb{Y}})}{\sum_{|\mathbb{Y}'|} \det(\boldsymbol{S}_{\mathbb{Y}'})}. \tag{23}$$

As such, our IP-DPP approach can effectively sample balanced subsets to rectify decision boundaries while preserving valuable information.

**Effective Sampling Strategy.** However, directly applying Eq. (23) for sampling entails significant computational costs. Drawing inspiration from prior studies [32, 33], we devise a novel and computationally efficient sampling strategy for our IP-DPP method. Specifically, given the symmetric stochastic matrix $\boldsymbol{S}$, its spectral decomposition yields orthonormal eigenvectors $\{\boldsymbol{v}_i\}_{i=1}^N$ with corresponding eigenvalues $\{\lambda_i\}_{i=1}^N$, such that:

$$\boldsymbol{S} = \sum_{i=1}^N \lambda_i \boldsymbol{v}_i \boldsymbol{v}_i^\top. \tag{24}$$

Let $\boldsymbol{e}_i \in \mathbb{R}^N$ denote the $i$-th standard basis vector, which contains a single 1 in its $i$-th entry and 0's elsewhere. $U(0,1)$ is the standard uniform distribution. Then, Algorithm 1 outlines an efficient sampling strategy for our IP-DPP approach.

Table 1: Experimental results on CIFAR-10-LT and CIFAR-100-LT datasets, with the best results shown in bold

| Methods | CIFAR-10-LT | | | | CIFAR-100-LT | | | |
|---|---|---|---|---|---|---|---|---|
| | Many-shot | Medium-shot | Few-shot | Overall | Many-shot | Medium-shot | Few-shot | Overall |
| Focal Loss | 86.3 | 60.6 | 46.3 | 69.2 | 71.1 | 43.9 | 10.5 | 43.5 |
| LDAM Loss | 85.8 | 64.8 | 51.9 | 71.5 | 71.4 | 44.5 | 11.7 | 44.1 |
| $\tau$-norm | 85.2 | 64.4 | 51.7 | 70.9 | 60.7 | 54.4 | 14.8 | 44.7 |
| RIDE | 86.2 | 63.6 | 56.1 | 73.4 | 73.1 | 47.6 | 16.4 | 47.2 |
| KCL | 83.7 | 63.8 | 53.6 | 71.7 | 72.3 | 46.1 | 14.8 | 45.8 |
| TSC | 81.5 | 71.9 | 56.3 | 71.9 | 71.3 | 43.9 | 10.5 | 43.5 |
| SBCL | 81.6 | 72.4 | 57.6 | 72.6 | 72.7 | 48.5 | 20.0 | 48.5 |
| OTmix | **87.9** | 67.8 | 47.3 | 73.8 | **73.1** | 48.0 | 19.1 | 48.1 |
| DisA | 86.1 | 68.3 | 50.3 | 73.6 | 72.4 | 49.3 | 21.9 | 49.2 |
| **Ours** | 82.0 | **76.3** | **67.2** | **76.4** | 62.4 | **59.7** | **31.9** | **52.4** |

# 5 Experimental Results

## 5.1 Experimental Setup

**Datasets.** We conduct experiments on four artificially induced or real-world long-tailed datasets: i) **CIFAR-10-LT** and ii) **CIFAR-100-LT**: we follow the setting in [3] by sampling long-tailed datasets respectively from the original CIFAR-10 and CIFAR-100 datasets; iii) **ImageNet-LT** [43]: a truncated version of ImageNet [12] with a total of $1,000$ classes; and iv) **iNaturalist 2018** [26]: a naturally long-tailed dataset containing $8,142$ species around the world. The imbalanced factor (IF) for CIFAR-10-LT and CIFAR-100-LT, if not specified, is set to $100$ (*i.e.*, $\frac{N_{\max}}{N_{\min}} = 100$).

**Compared Approaches.** We compare our approach to nine state-of-the-arts for long-tailed recognition: **Focal Loss** [38], **LDAM Loss** [3], $\tau$**-norm** [30], **RIDE** [59], **KCL** [34], **TSC** [35], **SBCL** [27], **OTmix** [15], and **DisA** [14].

**Metrics.** We evaluate long-tailed recognition performance using four metrics: **many-shot**, **medium-shot**, **few-shot**, and **overall** accuracies. Many-shot, medium-shot, and few-shot assess model performance in head, medium, and tail classes, respectively. All results are averaged over $5$ trials.

Additional experimental settings, including thresholds for defining the above metrics and hyperparameters, are provided in Appendix B.1 to conserve space.

## 5.2 Comparisons to State-of-the-Arts

**Small-Scale Datasets.** We first conduct experiments on two small-scale long-tailed datasets, *i.e.*, CIFAR-10-LT and CIFAR-100-LT, to compare our approach with nine counterparts mentioned in Section 5.1. Table 1 presents comparative results. On CIFAR-10-LT, our approach achieves the best overall accuracy of $76.4\%$, outperforming all counterparts by $2.6\%$ at least. This performance improvement can be attributed to two key aspects. First, our BNS approach effectively captures both instance-level and class-level semantics, facilitating the learning of high-quality representations and the creation of well-separated feature spaces, respectively. Second, our IP-DPP method addresses biased decision boundaries by sampling relatively balanced subsets while preserving valuable information. This can mitigate the majority-biased tendency while maintaining the model's overall performance. On the other hand, although our approach lags behind prior studies in many-shot accuracy, these methods consistently struggle with biased decision boundaries. They prioritize performance on head classes, resulting in significantly diminished accuracy for medium and tail classes. For instance, while our method falls short of OTmix by $5.9\%$ in many-shot accuracy, it surpasses OTmix with significantly higher medium-shot and few-shot accuracies, improving by $8.5\%$ and $19.9\%$, respectively. These results demonstrate that our approach effectively mitigates biased decision boundaries while maintaining the model's overall performance.

We observe similar trends on CIFAR-100-LT. First, our approach achieves the highest overall accuracy of $52.4\%$, surpassing prior state-of-the-art, *i.e.*, DisA, by a notable margin of $3.2\%$. Second, while our approach lags behind OTmix by $10.7\%$ in many-shot accuracy, it achieves improvements of $11.7\%$ in medium-shot accuracy and $12.8\%$ in few-shot accuracy, as well as an improvement of $4.3\%$ in overall accuracy. These results further confirm that our approach effectively mitigates majority-biased tendencies while preserving the model's overall performance.

Table 2: Experimental results on ImageNet-LT and iNaturalist 2018 datasets, with the best results highlighted in bold

| Methods | ImageNet-LT | | | | iNaturalist 2018 | | | |
|---|---|---|---|---|---|---|---|---|
| | Many-shot | Medium-shot | Few-shot | Overall | Many-shot | Medium-shot | Few-shot | Overall |
| Focal Loss | 51.4 | 41.2 | 16.0 | 41.7 | 61.2 | 62.7 | 64.4 | 63.2 |
| LDAM Loss | 55.0 | 46.4 | 16.7 | 45.7 | 65.1 | 66.8 | 61.7 | 64.6 |
| $\tau$-norm | 56.6 | 44.2 | 27.4 | 46.7 | 71.3 | 65.8 | 69.1 | 67.7 |
| RIDE | 56.7 | 46.4 | 25.7 | 47.6 | 67.5 | 68.6 | 69.3 | 68.8 |
| KCL | 55.0 | 42.6 | 25.4 | 45.0 | 61.2 | 62.7 | 64.4 | 63.2 |
| TSC | 57.1 | 45.2 | 29.3 | 47.6 | 66.4 | 65.7 | 64.0 | 65.1 |
| SBCL | 55.8 | 45.7 | 27.1 | 47.1 | **73.4** | 70.2 | 69.8 | 70.4 |
| OTmix | 50.9 | 46.0 | 25.7 | 45.1 | 70.1 | 70.9 | 68.6 | 69.9 |
| DisA | **61.0** | 47.0 | 25.3 | 49.4 | 70.7 | 70.8 | 68.4 | 69.8 |
| **Ours** | 59.7 | **50.8** | **32.4** | **51.7** | 72.7 | **72.9** | **75.7** | **74.0** |

**Large-Scale Datasets.** Next, we conduct experiments on ImageNet-LT and iNaturalist 2018 to assess the effectiveness of our approach on large-scale, long-tailed datasets. Table 2 provides the comprehensive results. We make two key observations. First, our approach achieves the highest accuracies on both ImageNet-LT (*i.e.*, 51.7%) and iNaturalist 2018 ( *i.e.*, 74.0%), outperforming all competing methods. For instance, on ImageNet-LT, our approach surpasses DisA, the best baseline method, by 2.3%. Similarly, on iNaturalist 2018, it outperforms SBCL, the best counterpart, by 3.6%. These results highlight the strong generalizability of our method to large-scale, long-tailed datasets. Moreover, while our approach lags behind some baseline methods in many-shot accuracy, it achieves the highest medium-shot and few-shot accuracies across both datasets. This is because prior methods result in majority-biased decision boundaries, which disproportionately favor head classes.

### 5.3 Evaluation on Representation Learning

**Quantitative Evaluation.** Next, we quantitatively evaluate the performance of our BNS method for representation learning. Specifically, we compare it against three state-of-the-art contrastive learning methods for long-tailed recognition: KCL, TSC, and SBCL. To assess the quality of the learned feature representations, we use linear probing accuracy, which involves fine-tuning a linear classifier on a pre-trained feature extractor with frozen weights.

Figures 1a and 1b illustrate linear probing accuracies on CIFAR-10-LT and CIFAR-100-LT, respectively. On CIFAR-10-LT, our approach achieves the best overall accuracy of 68.2% (see the pink bar), outperforming KCL, TSC, and SBCL by 7.2%, 4.4%, and 3.5%, respectively.

This is because maximizing the mutual information expressed in Eq. (3) is equivalent to minimizing the intra-class distance, which, in turn, enhances the quality of feature representations. Existing methods for long-tailed data often exhibit an undesirable bias toward head classes, leading to poor performance on tail classes, as evidenced by significant disparities between many-shot and few-shot accuracies: 52.7% for KCL, 45.1% for TSC, and 43.5% for SBCL. In contrast, our BNS method enjoys unbiased representation space, exhibiting similar perfor-

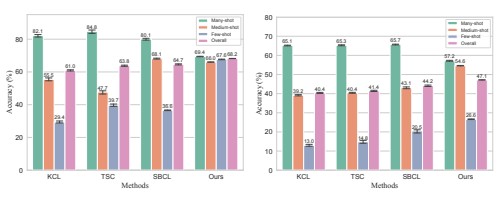

(a) CIFAR-10-LT      (b) CIFAR-100-LT

Figure 1: Linear probing accuracies on (a) CIFAR-10-LT and (b) CIFAR-100-LT datasets.

mance on many-shot and few-shot accuracies (*i.e.*, 69.4% vs. 67.6%). Similarly, our approach achieves the highest linear probing accuracy of 47.1% on CIFAR-100-LT, surpassing KCL, TSC, and SBCL by 6.7%, 5.7%, and 2.9%, respectively. Furthermore, our approach effectively mitigates the majority-biased tendency. For instance, compared to SBCL, our method achieves substantial improvements of 11.5% in medium-shot accuracy and 6.1% in few-shot accuracy, with only a modest 8.5% reduction in many-shot accuracy.

**Qualitative Evaluation.** To gain a deeper understanding of our BNS method's role in representation learning, we utilize t-SNE [56] to visualize the feature spaces learned by SBCL and our BNS approach. Figure 2 illustrates the feature spaces for the CIFAR-10 training and test sets. The training representation space learned by SBCL exhibits poor separation for medium and tail classes (see Figure 2a), leading to overlapping boundaries among these classes in the test representation space

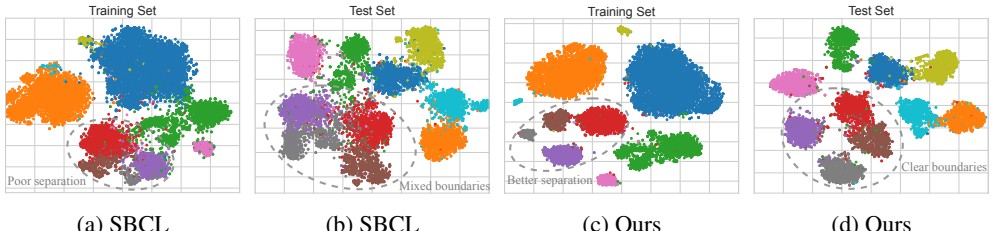

|                | (a) SBCL | (b) SBCL | (c) Ours | (d) Ours |

Figure 2: t-SNE visualization of CIFAR-10 feature space. (a) and (b): visual representation learned by SBCL, as well as (c) and (d): visual representation captured by our approach.

Table 3: Experimental results under various imbalanced factors (IF), with the best results shown in bold

| Methods | CIFAR-10-LT | | | | | CIFAT-100-LT | | | | |
|---|---|---|---|---|---|---|---|---|---|---|
| | IF=10 | IF=20 | IF=50 | IF=100 | IF=200 | IF=10 | IF=20 | IF=50 | IF=100 | IF=200 |
| TSC | 80.6 | 76.4 | 72.9 | 71.9 | 63.7 | 57.3 | 51.1 | 48.3 | 43.5 | 40.3 |
| SBCL | 80.5 | 78.6 | 74.1 | 72.6 | 64.3 | 58.3 | 51.8 | 50.9 | 48.5 | 41.4 |
| OTmix | 81.3 | 80.3 | 74.4 | 73.8 | 65.8 | 59.1 | 55.7 | 52.2 | 48.1 | 42.7 |
| DisA | 82.4 | 81.0 | 75.7 | 73.6 | 67.2 | 60.4 | 53.3 | 52.3 | 49.2 | 42.9 |
| Ours | **83.7** | **82.4** | **81.9** | **76.4** | **73.5** | **62.6** | **59.8** | **55.9** | **52.4** | **46.7** |

(see Figure 2b). In contrast, our approach achieves improved separation for medium and tail classes in the training representation space (see Figure 2c), leading to clear and well-defined boundaries for these classes in the test representation space (see Figure 2d). This improvement stems from our method's ability to capture class-level semantics, which promotes the development of well-separated representation spaces. The differences in decision boundaries between SBCL and our approach elucidate why SBCL achieves poor linear probing accuracies on medium and tail classes, whereas our approach demonstrates superior performance on these classes (see Figures 1a and 1b for details).

## 5.4 Performance Results under Various Imbalanced Factors

This section presents performance results under various imbalance factors (IF). Here, we consider five IF values ranging from 10 to 200. Table 3 shows comparative results on CIFAR-10-LT and CIFAR-100-LT, where we compare our approach with TSC, SBCL, OTmix, and DisA. On both datasets, our approach achieves the highest overall accuracies across all IF values. For instance, when the IF is set to 200, our method achieves an accuracy of 73.5% on CIFAR-10-LT and 46.7% on CIFAR-100-LT. Moreover, our method demonstrates greater robustness to large imbalance factors. For example, when the IF value increases from 10 to 200, our approach experiences a performance drop of 10.2% (*i.e.*, 83.7% vs. 73.5%) on CIFAR-10-LT, whereas baseline methods suffer larger decreases, with at least a 15.2% drop (see DisA, 82.4% vs. 67.2%).

Additional experimental results are provided in Appendices B.2–B.6. These include adaptability of our approach, applicability across different model architectures, evaluation of computational overhead, ablation studies, and hyperparameter sensitivity.

## 6 Conclusion

This work has addressed the challenging long-tailed data classification problem, by proposing a novel information-preservable two-stage learning approach. Our key contributions include: i) Balanced Negative Sampling (BNS), a new representation learning strategy that effectively captures both instance-level and class-level semantics, facilitating the creation of high-quality feature representations and well-separated feature spaces; and ii) Information-Preservable Determinantal Point Process (IP-DPP), a novel sampling technique designed to select mathematically informative instances, effectively rectifying majority-biased decision boundaries while maintaining the model's overall performance. As such, our approach achieves state-of-the-art performance across various long-tailed datasets by preserving the valuable information within the entire dataset, allowing it to consistently and decisively outperform its counterparts.

## Acknowledgments

This work was supported in part by NSF under Grants 2315613, 2438898, and 2348452. Any opinions and findings expressed in the paper are those of the authors and do not necessarily reflect the views of funding agencies.

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

## Outline

This document serves as supplementary material to the main paper, providing additional support in two key aspects. First, Appendix A presents detailed proofs for the theorems and lemmas presented in the main paper. Second, Appendix B includes additional experimental results that reinforce the findings. Third, Appendix C discusses both potential positive societal impacts and negative societal impacts of this work.

## A    Proofs of Theoretical Results

This section provides comprehensive proofs for the theoretical results discussed in the main paper. To enhance clarity and facilitate understanding of the mathematical details, we begin by restating the theoretical conclusions—such as the Theorems and Lemmas from the main paper—and then proceed with their detailed proofs.

### A.1    Detailed Derivation of Eq. (7)

This subsection supports Section 4.2 of the main paper by providing the detailed derivation of Eq. (7), as outlined below:

$$
\begin{aligned}
\log g(\boldsymbol{q}_i, \boldsymbol{v}_j \mid d = 1) &= \log \left[ \frac{p(\boldsymbol{q}_i, \boldsymbol{v}_{j,i}^+)}{p(\boldsymbol{q}_i, \boldsymbol{v}_{j,i}^+) + n \times p(\boldsymbol{q}_i)p(\boldsymbol{v}_j^-)} \right] = \log \left[ \frac{1}{1 + \frac{n \times p(\boldsymbol{q}_i)p(\boldsymbol{v}_j^-)}{p(\boldsymbol{q}_i, \boldsymbol{v}_{j,i}^+)}} \right] \\
&\leq \log \left[ \frac{1}{\frac{n \times p(\boldsymbol{q}_i)p(\boldsymbol{v}_j^-)}{p(\boldsymbol{q}_i, \boldsymbol{v}_{j,i}^+)}} \right] = \log \left[ \frac{p(\boldsymbol{q}_i, \boldsymbol{v}_{j,i}^+)}{p(\boldsymbol{q}_i)p(\boldsymbol{v}_j^-)} \cdot \frac{1}{n} \right] \\
&= \log \frac{p(\boldsymbol{q}_i, \boldsymbol{v}_{j,i}^+)}{p(\boldsymbol{q}_i)p(\boldsymbol{v}_j^-)} - \log n.
\end{aligned}
\tag{25}
$$

### A.2    Proof of Theorem 4.1

**Theorem A.1.** *(Intra-Class Distance Mutual Information Theorem) Let $\mathbb{X}_Q^c$ and $\mathbb{X}_V^c$ be two sets of images with the same label $c$, obtained by different data augmentation techniques. Given a feature extractor $f_{\boldsymbol{\theta}}(\cdot)$, we define $\boldsymbol{Q}^c$ and $\boldsymbol{V}^c$ as the representation spaces for $\mathbb{X}_Q^c$ and $\mathbb{X}_V^c$, respectively. Then, any pair of $\boldsymbol{q}_i^c \in \boldsymbol{Q}^c$ and $\boldsymbol{v}_j^c \in \boldsymbol{V}^c$ is a positive pair. Let $MI(\cdot)$ and $D(\cdot)$ respectively denote the mutual information and a distance metric, we have:*

$$
\max MI(\boldsymbol{Q}^c, \boldsymbol{V}^c) \propto \min D(\boldsymbol{Q}^c, \boldsymbol{V}^c),
\tag{26}
$$

*where $D(\boldsymbol{Q}^c, \boldsymbol{V}^c)$ can be considered as the intra-class distance because they have the same label.*

*Proof.* According to Variation of Information (VI) [46], we have:

$$
\begin{aligned}
D(\boldsymbol{Q}^c, \boldsymbol{V}^c) &= H(\boldsymbol{Q}^c, \boldsymbol{V}^c) - MI(\boldsymbol{Q}^c, \boldsymbol{V}^c) \\
&= H(\boldsymbol{Q}^c) + H(\boldsymbol{V}^c) - 2MI(\boldsymbol{Q}^c, \boldsymbol{V}^c).
\end{aligned}
\tag{27}
$$

Here, $H(\cdot)$ represents the entropy. For the given $\boldsymbol{Q}^c$ and $\boldsymbol{V}^c$, $H(\boldsymbol{Q}^c)$ and $H(\boldsymbol{V}^c)$ are two constants. Therefore, we have $\max MI(\boldsymbol{Q}^c, \boldsymbol{V}^c) \propto \min D(\boldsymbol{Q}^c, \boldsymbol{V}^c)$. That is, maximizing the mutual information between $\boldsymbol{Q}^c$ and $\boldsymbol{V}^c$ is proportional to minimizing their intra-class distance. □

### A.3    Proof of Lemma 4.4

**Lemma A.2.** *(Positive Semi-definiteness) Let $p(i) = p(y_i|\boldsymbol{x}_i)$ represent the probability of $y_i$ given $\boldsymbol{x}_i$. $\boldsymbol{S} \in \mathbb{R}^{N \times N}$ is a symmetric stochastic matrix, where each row (or column) sums to 1. Then, we have:*

$$
\boldsymbol{v}^\top \boldsymbol{S} \boldsymbol{v} \geq 0, \quad \forall \boldsymbol{v} \in \mathbb{R}^N.
\tag{28}
$$

*In other words, $\boldsymbol{S}$ is positive semi-definite.*

*Proof.* According to the basic decomposition of symmetric matrices, we have $S = Q\Lambda Q^\top$, where $Q$ is an orthogonal matrix $QQ^\top = I$, and $\Lambda$ is a diagonal matrix, with diagonal entities containing all eigenvalues of $S$. Given any vector $v \in \mathbb{R}^N$, we let $c = Q^\top v$. In other words, $v = Qc$. Then, we arrive at:

$$v^\top S v = c^\top Q^\top Q \Lambda Q^\top Q c = \Lambda c^\top c = \sum_{i=1}^{N} \lambda_i c^\top c, \tag{29}$$

where $\{\lambda\}_{i=1}^N$ represents the eigenvalues of $S$, and we have $c^\top c \geq 0$. Let $\text{Tr}(\cdot)$ denote a matrix's trace. Then, regarding the trace of $S$, we have:

$$\text{Tr}(S) = \text{Tr}(Q\Lambda Q^\top) = \text{Tr}(\Lambda Q Q^\top) = \text{Tr}(\Lambda) = \sum_{i=1}^{n} \lambda_i \geq 0. \tag{30}$$

Combining Eq. (29) and Eq. (30), we obtain $v^\top S v \geq 0$ for any vector $v \in \mathbb{R}^N$. Therefore, $S$ is positive semi-definite. $\qquad\square$

## A.4 Proof of Lemma 4.5

**Lemma A.3.** *(Bounds on Eigenvalues) Let $\{\lambda_i\}_{i=1}^N$ be the eigenvalues of the symmetric stochastic matrix $S \in \mathbb{R}^{N \times N}$, we have:*

$$0 \leq \lambda_i \leq 1, \quad \forall \lambda_i. \tag{31}$$

*Proof.* Since $S \in \mathbb{R}^{N \times N}$ is a symmetric stochastic matrix, the following conditions always hold:

$$S_{i,j} \geq 0, \quad \forall\, 1 \leq i, j \leq N; \tag{32a}$$

$$\sum_{j=1}^{N} S_{i,j} = 1, \quad \forall\, 1 \leq i \leq N. \tag{32b}$$

Let $\lambda$ be an eigenvalue of $S$, with its corresponding eigenvector denoted as $v = [v_1, v_2, \cdots, v_N]^\top$. Then, we obtain:

$$Sv = \lambda v. \tag{33}$$

Suppose $v_k \in v$ has the largest absolute value, *i.e.*, $|v_k| \geq |v_i|$ for all $1 \leq i \leq N$. According to Eq. (33), for the $k$-th row of $S$, we have:

$$S_{k,1} v_1 + S_{k,2} v_2 + \cdots + S_{k,N} v_N = \lambda v_k. \tag{34}$$

Combining Eq. (32a), Eq. (32b), and Eq. (34), we arrive at:

$$\begin{aligned}
|\lambda| \cdot |v_k| = |\lambda v_k| &= |S_{k,1} v_1 + S_{k,2} v_2 + \cdots + S_{k,N} v_N| \\
&\leq |S_{k,1} v_1| + |S_{k,2} v_2| + \cdots + |S_{k,N} v_N| \\
&\leq |S_{k,1} v_k| + |S_{k,2} v_k| + \cdots + |S_{k,N} v_k| \\
&= |v_k| \sum_{j=1}^{N} S_{k,j} \\
&= |v_k|.
\end{aligned} \tag{35}$$

According to Eq. (35), we obtain $|\lambda| \cdot |v_k| \leq |v_k|$. Therefore, for all $1 \leq i \leq N$, we always have $|\lambda_i| \leq 1$. Meanwhile, according to Lemma A.2, we obtain $\lambda_i \geq 0$ for all $1 \leq i \leq N$. Finally, combining both scenarios, we arrive at the following:

$$0 \leq \lambda_i \leq 1, \quad \forall \lambda_i. \tag{36}$$

$\qquad\square$

## A.5 Proof of Theorem 4.6

**Theorem A.4.** *(Bounded Determinant Probability Measure) Let $\mathbb{X}$ be a ground set with $N$ items, and let $\boldsymbol{S} \in \mathbb{R}^{N \times N}$ denote its similarity matrix. Here, $\boldsymbol{S}$ is positive semidefinite and satisfies $0 \preceq \boldsymbol{S} \preceq \boldsymbol{I}$, where $\boldsymbol{I}$ is the $N \times N$ identity matrix. For any subset $\mathbb{Y} \subseteq \mathbb{X}$, let $\boldsymbol{S}_{\mathbb{Y}}$ denote the principal submatrix of $\boldsymbol{S}$ corresponding to $\mathbb{Y}$. Then, the following holds:*

$$0 \leq \frac{\det(\boldsymbol{S}_{\mathbb{Y}})}{\det(\boldsymbol{S} + \boldsymbol{I})} \leq 1. \tag{37}$$

*In other words, the value $\mathcal{P}_{\boldsymbol{S}}(\mathbb{Y}) = \frac{\det(\boldsymbol{S}_{\mathbb{Y}})}{\det(\boldsymbol{S}+\boldsymbol{I})}$ defines a valid probability measure.*

*Proof.* Since $\boldsymbol{S}$ is positive semidefinite, both the principal submatrix $\boldsymbol{S}_{\mathbb{Y}}$ and the matrix $\boldsymbol{S} + \boldsymbol{I}$ are also positive semidefinite. If a matrix is positive semidefinite, all its eigenvalues are non-negative. Then, we have $det(\boldsymbol{S}_{\mathbb{Y}}) \geq 0$ and $det(\boldsymbol{S} + \boldsymbol{I}) > 0$. Therefore, $\frac{\det(\boldsymbol{S}_{\mathbb{Y}})}{\det(\boldsymbol{S}+\boldsymbol{I})} \geq 0$ always holds.

To establish that $\frac{\det(\boldsymbol{S}_{\mathbb{Y}})}{\det(\boldsymbol{S}+\boldsymbol{I})} \leq 1$, we aim to prove the key identity:

$$\sum_{\mathbb{Y} \subseteq \mathbb{X}} \det(\boldsymbol{S}_{\mathbb{Y}}) = \det(\boldsymbol{S} + \boldsymbol{I}_{\mathbb{Y}}). \tag{38}$$

For any $\boldsymbol{A} \subseteq \mathbb{X}$, Eq. (38) is a special case of the following equation:

$$\sum_{\boldsymbol{A} \subseteq \mathbb{Y} \subseteq \mathbb{X}} \det(\boldsymbol{S}_{\mathbb{Y}}) = \det(\boldsymbol{S} + \boldsymbol{I}_{\bar{\boldsymbol{A}}}), \tag{39}$$

where $\boldsymbol{I}_{\bar{\boldsymbol{A}}}$ is a diagonal matrix with the following properties: entries are 1 for indices corresponding to elements in $\bar{\boldsymbol{A}} = \mathbb{X} - \boldsymbol{A}$, and entries are 0 for all other positions. Therefore, if Eq. (39) is satisfied, it follows that $\frac{\det(\boldsymbol{S}_{\mathbb{Y}})}{\det(\boldsymbol{S}+\boldsymbol{I})} \leq 1$. This result is motivated by Theorem 2.1 in the prior work [33].

Specifically, for the case $\boldsymbol{A} = \mathbb{X}$, Eq. (39) always holds. For the case where $\boldsymbol{A} \subset \mathbb{X}$, we assume Eq. (39) holds whenever $\bar{\boldsymbol{A}}$ has cardinality less than $k$ (where $k > 0$), *i.e.*, $|\bar{\boldsymbol{A}}| = k$. Let $i$ be an arbitrary element of $\bar{\boldsymbol{A}}$, so $i \in \bar{\boldsymbol{A}}$. By partitioning the ground set $\mathbb{X}$ into $\{i\}$ and $\mathbb{X} - \{i\}$, we can decompose the problem as follows:

$$\boldsymbol{S} + \boldsymbol{I}_{\bar{\boldsymbol{A}}} = \begin{pmatrix} S_{ii} + 1 & \boldsymbol{S}_{i\bar{i}} \\ \boldsymbol{S}_{\bar{i}i} & \boldsymbol{S}_{\mathbb{X}-\{i\}} + \boldsymbol{I}_{\mathbb{X}-\{i\}-\boldsymbol{A}} \cdot \end{pmatrix} \tag{40}$$

Here, $\boldsymbol{S}_{\bar{i}i}$ denote the subcolumn of the $i$-th column of $\boldsymbol{S}$, restricted to rows corresponding to elements in $\bar{i}$. Similarly, $\boldsymbol{S}_{i\bar{i}}$ represents the corresponding subcolumn but transposed. By leveraging the multilinearity property of determinants, we observe:

$$\begin{aligned} \det(\boldsymbol{S} + \boldsymbol{I}_{\bar{\boldsymbol{A}}}) &= \begin{vmatrix} \boldsymbol{S}_{ii} & \boldsymbol{S}_{i\bar{i}} \\ \boldsymbol{S}_{\bar{i}i} & \boldsymbol{S}_{\mathbb{X}-\{i\}} + \boldsymbol{I}_{\mathbb{X}-\{i\}-\boldsymbol{A}} \end{vmatrix} + \begin{vmatrix} 1 & 0 \\ \boldsymbol{S}_{\bar{i}i} & \boldsymbol{S}_{\mathbb{X}-\{i\}} + \boldsymbol{I}_{\mathbb{X}-\{i\}-\boldsymbol{A}} \end{vmatrix} \\ &= \det\left(\boldsymbol{S} + \boldsymbol{I}_{\overline{\boldsymbol{A} \cup \{i\}}}\right) + \det\left(\boldsymbol{S}_{\mathbb{X}-\{i\}} + \boldsymbol{I}_{\mathbb{X}-\{i\}-\boldsymbol{A}}\right). \end{aligned} \tag{41}$$

By applying the inductive hypothesis to each term separately, we obtain:

$$\begin{aligned} \det(\boldsymbol{S} + \boldsymbol{I}_{\bar{\boldsymbol{A}}}) &= \sum_{\boldsymbol{A} \cup \{i\} \subseteq \mathbb{Y} \subseteq \mathbb{X}} \det(\boldsymbol{S}_{\mathbb{Y}}) + \sum_{\boldsymbol{A} \subseteq \mathbb{Y} \subseteq \mathbb{X}-\{i\}} \det(\boldsymbol{S}_{\mathbb{Y}}) \\ &= \sum_{\boldsymbol{A} \subseteq \mathbb{Y} \subseteq \mathbb{X}} \det(\boldsymbol{S}_{\mathbb{Y}}). \end{aligned} \tag{42}$$

We observe that each subset $\mathbb{Y}$ falls into exactly one of two mutually exclusive categories: either $\mathbb{Y}$ contains the element $i$ (contributing to the first sum) or $\mathbb{Y}$ does not contain $i$ (contributing to the second sum). According to Eq. (38), Eq. (39), and Eq. (42), we have $\frac{\det(\boldsymbol{S}_{\mathbb{Y}})}{\det(\boldsymbol{S}+\boldsymbol{I})} \leq 1$.

Finally, combining both scenarios, we arrive at $0 \leq \frac{\det(\boldsymbol{S}_{\mathbb{Y}})}{\det(\boldsymbol{S}+\boldsymbol{I})} \leq 1$.

$\square$

---

**Algorithm 2** Efficient Sampling Algorithm for a Standard DPP

---

1: **Input:** a ground set $\mathbb{X} = \{\boldsymbol{x}_i\}_{i=1}^N$ and its symmetric stochastic matrix $\boldsymbol{S}$
2: **Initialize:** standard basis vectors $\{\boldsymbol{e}_i\}_{i=1}^N$ and pairs of orthonormal eigenvalues and eigenvectors $\{(\lambda_i, \boldsymbol{v}_i)\}_{i=1}^N$ for $\boldsymbol{S}$
3: $\boldsymbol{V} \leftarrow \emptyset$
4: **for** $i = 1, 2, \cdots, N$ **do**
5:    **if** $u \sim U(0,1) < \frac{\lambda_i}{\lambda_i+1}$ **then**
6:       $\boldsymbol{V} \leftarrow \boldsymbol{V} \cup \{\boldsymbol{v}_i\}$
7:    **end if**
8: **end for**
9: $\mathbb{Y} \leftarrow \emptyset$
10: **while** $|\boldsymbol{V}| > 0$ **do**
11:    **for** $i = 1, 2, \cdots, N$ **do**
12:       $p(i) \leftarrow \frac{1}{|\boldsymbol{V}|} \sum_{\boldsymbol{v} \in \boldsymbol{V}} (\boldsymbol{v}^\top \boldsymbol{e}_i)^2$
13:    **end for**
14:    $i^* \leftarrow \arg\max_i p(i)$
15:    $\mathbb{Y} \leftarrow \mathbb{Y} \cup \{\boldsymbol{x}_{i^*}\}$
16:    $\boldsymbol{V} \leftarrow \boldsymbol{V}_\perp$ // Update $\boldsymbol{V}$ to an orthonormal basis for the subspace orthogonal to $\boldsymbol{e}_{i^*}$
17: **end while**
18: **Return:** a subset $\mathbb{Y}$

---

### A.6 Detailed Derivation of Eq. (22)

This subsection supports Section 4.3 by providing the detailed derivation of Eq. (22). Given that $\boldsymbol{A} = \{i, j\}$ is a subset with two elements, we have $N = 2$. Then, the derivation is as follows:

$$
\begin{aligned}
\mathcal{P}_{\boldsymbol{S}}(\boldsymbol{A}) = \frac{\det(\boldsymbol{S}_{\boldsymbol{A}})}{\det(\boldsymbol{S}+\boldsymbol{I})} \propto \det(\boldsymbol{S}_{\boldsymbol{A}}) &= \begin{vmatrix} 1 - \frac{p(i) \cdot p(j)}{N} & \frac{p(i) \cdot p(j)}{N} \\ \frac{p(i) \cdot p(j)}{N} & 1 - \frac{p(i) \cdot p(j)}{N} \end{vmatrix} = \begin{vmatrix} 1 - \frac{p(i) \cdot p(j)}{2} & \frac{p(i) \cdot p(j)}{2} \\ \frac{p(i) \cdot p(j)}{2} & 1 - \frac{p(i) \cdot p(j)}{2} \end{vmatrix} \\
&= (1 - \frac{p(i) \cdot p(j)}{2})^2 - (\frac{p(i) \cdot p(j)}{2})^2 \\
&= 1 - p(i) \cdot p(j).
\end{aligned}
\tag{43}
$$

### A.7 Expected Sample Size of a Determinantal Point Process (DPP)

This subsection complements the main paper by presenting the theorem and its corresponding proof for the expected sample size of a DPP. To facilitate understanding of the theoretical results, Algorithm 2 presents an efficient sampling method for a standard DPP, *i.e.*, DPP without fixed sample size $k$.

**Theorem A.5.** *(Expected Sample Size of a DPP) Let $\mathbb{X}$ be a ground set containing $N$ elements. For any subset $\mathbb{Y} \subseteq \mathbb{X}$ sampled by a Determinantal Point Process (DPP), the expected size of $\mathbb{Y}$ is given by:*

$$
\mathbb{E}[|\mathbb{Y}|] = N(1 - \ln 2).
\tag{44}
$$

*Proof.* The size of the sampled subset $|\mathbb{Y}|$ from Algorithm 2 is determined by the cardinality of the selected eigenvector set $|\boldsymbol{V}|$. This cardinality follows a Poisson-binomial distribution (see Line 5 in Algorithm 2 for details), where each of the $N$ independent Bernoulli trials succeeds with probability $p_i = \frac{\lambda_i}{\lambda_i+1}$, corresponding to the $i$-th eigenvalue $\lambda_i$ of the kernel matrix. Thus, $|\mathbb{Y}| \sim \sum_{i=1}^N \text{Bernoulli}(p_i)$.

According to Lemma A.4, we have $0 \leq \lambda_i \leq 1, \forall \lambda_i$. Now, we assume that $\lambda_i$ is a random variable distributed over the interval $[0, 1]$. Denote this random variable as $\lambda$. The expected value is given by:

$$
\mathbb{E}\left[\frac{\lambda}{\lambda+1}\right] = \int_0^1 \frac{\lambda}{\lambda+1} f(\lambda) d\lambda,
\tag{45}
$$

where $f(\lambda)$ is the probability density function (PDF) of $\lambda$. Without loss of generality, suppose $\lambda$ is uniformly distributed over $[0, 1]$, the PDF is $f(\lambda) = 1$ for $\lambda \in [0, 1]$. The expected value becomes:

$$\mathbb{E}\left[\frac{\lambda}{\lambda + 1}\right] = \int_0^1 \frac{\lambda}{\lambda + 1} d\lambda. \tag{46}$$

Let $u = \lambda + 1$, so $du = d\lambda$ and when $\lambda = 0$, $u = 1$; when $\lambda = 1$, $u = 2$. The integral becomes:

$$\begin{aligned}
\int_0^1 \frac{\lambda}{\lambda + 1} d\lambda = \int_1^2 \frac{u - 1}{u} du &= \int_1^2 \left(1 - \frac{1}{u}\right) du \\
&= \int_1^2 1 du - \int_1^2 \frac{1}{u} du = [u]_1^2 - [\ln u]_1^2 \\
&= 1 - \ln 2.
\end{aligned} \tag{47}$$

Therefore, we have $\bar{p}_i = \mathbb{E}\left[\frac{\lambda_i}{\lambda_i + 1}\right] = 1 - \ln 2$. Finally, the expected size of the set $\mathbb{Y}$ can be approximated as follows:

$$\mathbb{E}\left[|\mathbb{Y}|\right] = \sum_{i=1}^N \bar{p}_i = \sum_{i=1}^N (1 - \ln 2) = N(1 - \ln 2). \tag{48}$$

$\square$

Notably, we assume a uniform distribution of eigenvalues to simplify the above proof. Although this assumption is idealized, the empirical results (see Appendix B.7) on the subset sample size after a DPP demonstrate that it does not substantially distort the practical behavior of the process.

## B  Supplementary Experimental Results

### B.1  Additional Experimental Settings

**Metric Threshold.** We define thresholds for many-shot, medium-shot, and few-shot accuracies. Specifically, for CIFAR-10-LT, many-shot refers to classes with more than 500 images, medium-shot to classes with 200 to 500 images, and few-shot to classes with fewer than 200 images. For other datasets, many-shot refers to classes with more than 100 images, medium-shot to those with 20 to 100 images, and few-shot to those with fewer than 20 images.

**Hyperparameters.** ResNet-18, ResNet-34, and ResNet-50 [24] are used for CIFAR-10-LT, CIFAR-100-LT, and ImageNet-LT (or iNaturalist 2018), respectively. In the first stage, the feature extractor is trained using the AdamW [45] optimizer with $\beta_1 = 0.9$, $\beta_2 = 0.95$, and a weight decay of $0.05$. The training process consists of $1,000$ epochs, including $20$ warm-up epochs, with a batch size of $1,024$. A cosine learning rate decay schedule [44] is applied, starting with a base learning rate of $10e - 3$ and incorporating a layer-wise learning rate decay [8] of $0.75$. Data augmentation strategies, including random cropping, random color distortions, and random Gaussian blur, are used, followed by those reported in SimCLR [5]. The number of additional positive pairs $m$ was set to 6 by default. In the second stage, the pre-trained feature extractor is fine-tuned for 100 epochs, including 5 warm-up epochs, with a batch size of 64. The sample size $k$ is set to $10N_C$, where $N_C$ represents the sample size of the smallest class. All experiments were conducted on a workstation equipped with an RTX 4090 GPU.

### B.2  Adaptability of Our Two-Stage Learning Approach

This work introduces a novel two-stage learning approach, incorporating Balanced Negative Sampling (BNS) for representation learning in the first stage and Information-Preservable Determinantal Point Process (IP-DPP) for sampling the balanced training set in the second stage. In this subsection, we conduct experiments to demonstrate that our BNS and IP-DPP approaches can be easily adapted by existing studies. Specifically, BNS is combined with re-weighting methods, *i.e.*, Focal Loss and LDAM Loss, while IP-DPP is combined with prior representation learning approaches, *i.e.*, KCL, TSC, and SBCL. Table 4 presents the experimental results on CIFAR-10-LT and CIFAR-100-LT, where the performance outcomes of existing methods are also included for a better comparison.

Table 4: Experimental results on CIFAR-10-LT and CIFAR-100-LT, with the best results shown in bold

| Methods | CIFAR-10-LT | | | | CIFAR-100-LT | | | |
|---|---|---|---|---|---|---|---|---|
| | Many-shot | Medium-shot | Few-shot | Overall | Many-shot | Medium-shot | Few-shot | Overall |
| Focal Loss | **86.3** | 60.6 | 46.3 | 69.2 | 71.1 | 43.9 | 10.5 | 43.5 |
| LDAM Loss | 85.8 | 64.8 | 51.9 | 71.5 | 71.4 | 44.5 | 11.7 | 44.1 |
| KCL | 83.7 | 63.8 | 53.6 | 71.7 | 72.3 | 46.1 | 14.8 | 45.8 |
| TSC | 81.5 | 71.9 | 56.3 | 71.9 | 71.3 | 43.9 | 10.5 | 43.3 |
| SBCL | 81.6 | 72.4 | 57.6 | 72.6 | **72.7** | 48.5 | 20.0 | 48.5 |
| BNS + Focal Loss | 82.2 | 73.6 | 57.1 | 72.9 | 63.6 | 57.7 | 27.8 | 50.8 |
| BNS + LDAM Loss | 82.6 | 71.8 | 61.9 | 74.2 | 63.3 | **58.7** | 27.8 | 51.0 |
| KCL + IP-DPP | 79.2 | 75.7 | 66.6 | 74.7 | 64.2 | 58.3 | 26.7 | 50.8 |
| TSC + IP-DPP | 78.9 | **76.8** | 63.4 | 73.8 | 63.2 | 58.3 | **30.1** | **51.5** |
| SBCL + IP-DPP | 80.0 | 73.5 | **66.8** | **74.8** | 62.7 | 57.8 | 29.7 | 51.2 |

Table 5: Experimental results on ImageNet-LT and iNaturalist 2018 using various model architectures. Best results are highlighted in bold

| Models | ResNet-50 | ViT-Base | DeiT-Base | Swin-Base |
|---|---|---|---|---|
| ImageNet-LT | **51.7** | 50.1 | 50.8 | 51.2 |
| iNaturalist 2018 | 74.0 | 72.6 | 74.3 | **74.6** |

Table 6: Comparative analysis of computational overhead. The sampling time, measured in seconds, is reported.

| Datasets | CIFAR-10-LT | CIFAR-100-LT | ImageNet-LT |
|---|---|---|---|
| Random | 12.4 | 14.6 | 118.2 |
| IP-DPP (w/o efficient sampling) | 19.0 | 36.8 | 399.8 |
| IP-DPP | 13.2 | 15.8 | 144.0 |

Our approach effectively enhances the performance of existing methods. For instance, combining SBCL with IP-DPP (*i.e.*, "SBCL + IP-DPP") achieves the highest overall accuracy of 74.8% on CIFAR-10-LT. Similarly, integrating TSC with IP-DPP (*i.e.*, "TSC + IP-DPP") yields the best overall accuracy of 51.5% on CIFAR-100-LT. Furthermore, integrating our approach with existing methods consistently improves their performance compared to the original methods. For instance, combining BNS with Focal Loss (*i.e.*, "BNS + Focal Loss") outperforms "Focal Loss" by 3.7% (*i.e.*, 72.9% vs. 69.2%) on CIFAR-10-LT and by 7.3% (*i.e.*, 50.8% vs. 43.5%) on CIFAR-100-LT. These results demonstrate that our BNS and IP-DPP approaches can be effectively integrated into other methods to enhance performance in imbalanced classification tasks.

### B.3 Applicability Across Various Model Architectures

In this subsection, we conduct experiments to evaluate whether our approach can be applied effectively across different model architectures. Table 5 presents the results on ImageNet-LT and iNaturalist 2018 using four architectures: ResNet-50 [24], ViT-Base [13], DeiT-Base [55], and Swin-Base [41]. Our approach demonstrates consistent performance across different model architectures. For instance, on ImageNet-LT, it achieves 51.7% accuracy using ResNet and 51.2% accuracy using Swin-Base. Similarly, on iNaturalist 2018, ResNet-50 achieves an overall accuracy of 74.0%, while Swin-Base achieves a slightly higher accuracy of 74.6%. These results highlight the generalizability of our approach to a variety of model architectures, confirming its robustness and flexibility.

### B.4 Evaluation of Computational Overhead

This subsection supports the main paper by presenting an evaluation of the computational overhead introduced by our IP-DPP method. We first examine the additional sampling time incurred by IP-DPP. Specifically, we evaluate three scenarios: the training set sampled using Random Undersampling [21], IP-DPP without the efficient sampling strategy, and IP-DPP with the proposed strategy. Table 6 provides a comparative analysis of these methods across three long-tailed datasets: CIFAR-10-LT, CIFAR-100-LT, and ImageNet-LT. The results reveal two key observations. First, while our IP-DPP approach introduces additional computational overhead compared to Random Undersampling, the

Table 7: Analysis of computational overhead. The total sampling time and training time are reported in seconds. The baseline method does not involve additional sampling, so its sampling time costs are not applicable.

| Datasets | CIFAR-10-LT | | CIFAR-100-LT | |
|---|---|---|---|---|
| | Sampling Time | Training Time | Sampling Time | Training Time |
| Baseline | — | 537.2 | — | 781.6 |
| IP-DPP | 132.0 | 484.4 | 158.0 | 491.8 |

Table 8: Linear probing accuracy on CIFAR-10-LT and CIFAR-100-LT, with the best results highlighted in bold

| Datasets | CIFAR-10-LT | | | | CIFAR-100-LT | | | |
|---|---|---|---|---|---|---|---|---|
| | Many-shot | Medium-shot | Few-shot | Overall | Many-shot | Medium-shot | Few-shot | Overall |
| Baseline | 66.7 | 63.5 | 34.8 | 56.5 | 55.1 | 50.2 | 10.1 | 39.9 |
| BNS (w/o additional positive pairs) | 68.3 | 64.7 | 43.3 | 60.1 | 56.7 | 53.1 | 20.9 | 44.7 |
| BNS | **69.4** | **66.0** | **67.6** | **68.2** | **57.2** | **54.6** | **26.6** | **47.1** |

difference is minimal, with IP-DPP being only $0.8$ seconds slower on CIFAR-10-LT, $0.8$ seconds slower on CIFAR-100-LT, and $25.8$ seconds slower on ImageNet-LT. Second, the effective sampling strategy of IP-DPP significantly reduces computational overhead. Compared to its counterparts without the efficient sampling strategy, IP-DPP achieves speedups of $1.4$x on CIFAR-10-LT, $2.3$x on CIFAR-100-LT, and $2.8$x on ImageNet-LT.

Next, we conduct experiments to evaluate the impact of our IP-DPP approach on total training time. Table 7 summarizes the total training time for CIFAR-10-LT and CIFAR-100-LT over 100 epochs, with the total training cost of Focal Loss serving as the baseline. In practice, the training set is re-sampled using our IP-DPP approach every 10 epochs, a strategy designed to mitigate overfitting. Therefore, Table 7 reports the cost of performing ten sampling iterations as the total sampling time. Although our IP-DPP approach incurs additional sampling time, it results in a lower total training time compared to the baseline method. This reduction is attributed to the significant decrease in the sample size of the training set after using IP-DPP.

## B.5 Comprehensive Ablation Studies

**Ablation Studies on Our BNS Approach.** We conduct experiments on long-tailed datasets to evaluate the effectiveness of our BNS approach in representation learning. For this, we use SimCLR [5], a conventional representation learning method, as the baseline. We examine two variants of our approach: one without additional positive pairs and another with them. Table 8 presents the linear probing accuracies on CIFAR-10-LT and CIFAR-100-LT datasets. We make three key observations. First, the conventional method struggles to learn high-quality feature spaces for long-tailed datasets, leading to poor linear probing accuracy in tail classes, *e.g.*, $34.8\%$ on CIFAR-10-LT and $10.1\%$ on CIFAR-100-LT. Second, both variants of our BNS approach significantly outperform the baseline method. This improvement is attributed to their ability to effectively capture instance-level semantics, thereby enhancing the quality of feature representations. Third, compared to its variant without additional positive pairs, our BNS approach with additional positive pairs achieves superior linear probing accuracy, particularly in tail classes (*e.g.*, $67.6\%$ vs. $43.3\%$ on CIFAR-10-LT). This improvement is due to the inclusion of multiple positive pairs, which enables our BNS method to capture class-level semantics, thereby promoting a well-separated feature space.

**Ablation Studies on Our IP-DPP Approach.** We conduct experiments to evaluate the necessity and significance of the innovative designs within our IP-DPP approach. In these experiments, Random Undersampling [21] serves as the baseline method. For our IP-DPP approach, we examine its variants: IP-DPP without the symmetric stochastic matrix, IP-DPP without the fixed sample size, and the complete IP-DPP method. Table 9 presents the comparative results on CIFAR-10-LT and CIFAR-100-LT. We have three observations. First, compared to the baseline method, all IP-DPP variants achieve better overall accuracies on both datasets. This is because the baseline method incurs significant information loss due to its random undersampling strategy. Second, removing either the symmetric stochastic matrix or the fixed sample size results in significant performance degradation.

Table 9: Ablation studies on our IP-DPP technique, with the best results shown in bold

| Methods | CIFAR-10-LT | | | | CIFAR-100-LT | | | |
|---|---|---|---|---|---|---|---|---|
| | Many-shot | Medium-shot | Few-shot | Overall | Many-shot | Medium-shot | Few-shot | Overall |
| Baseline | 75.5 | 72.8 | 61.9 | 70.8 | 56.7 | 56.4 | 24.1 | 47.8 |
| IP-DPP (w/o stochastic matrix) | 81.4 | 74.4 | 57.1 | 72.7 | 61.9 | 57.7 | 23.0 | 48.7 |
| IP-DPP (w/o fixed sample size) | **82.7** | 74.7 | 63.8 | 75.4 | **65.5** | 57.0 | 26.7 | 50.8 |
| IP-DPP | 82.0 | **76.3** | **67.2** | **76.4** | 62.4 | **59.7** | **31.9** | **52.4** |

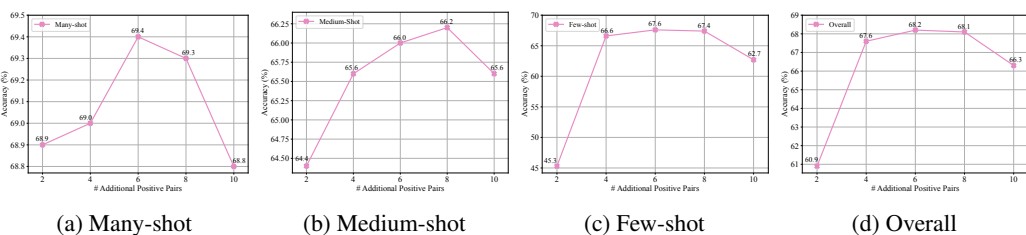

| (a) Many-shot | (b) Medium-shot | (c) Few-shot | (d) Overall |
|---|---|---|---|

Figure 3: Linear probing accuracy on CIFAR-10-LT using different numbers of additional positive pairs (*i.e.*, $m$).

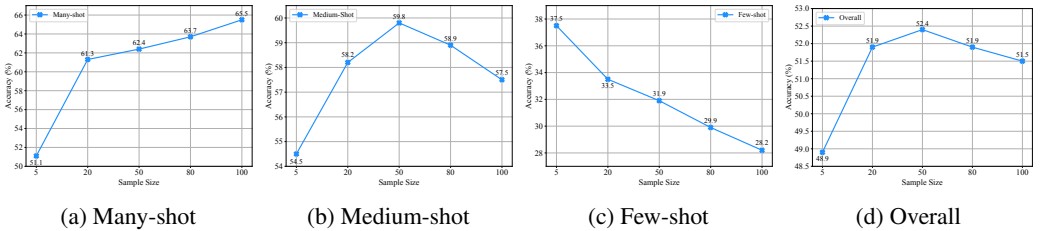

| (a) Many-shot | (b) Medium-shot | (c) Few-shot | (d) Overall |
|---|---|---|---|

Figure 4: Impact of the fixed sample size (*i.e.*, $k$) on imbalanced classification, where many-shot, medium-shot, few-shot, and overall accuracies on CIFAR-100 are reported.

This is because the symmetric stochastic matrix is essential for preserving mathematically informative samples, while the fixed sample size plays a crucial role in maintaining data balance after applying IP-DPP. These statistical results validate the importance of the novel designs within IP-DPP in rectifying majority-biased decision boundaries while preserving the model's overall performance.

## B.6 Hyperparameter Sensitivity

**Effects of Additional Positive Pairs.** Here, we investigate the impact of the number of positive pairs in our BNS approach on representation learning. Notably, given any anchor image, we have at least one positive pair, *i.e.*, two augmented images from the anchor image. Here, we conduct experiments by gradually increasing the number of additional positive pairs, *i.e.*, $m$ in Eq. (12). Figure 3 illustrate the linear probing accuracy on CIFAR-10-LT, where $m$ is gradually increased from 2 to 10. As shown in Figure 3d, when the value of $m$ is small ($m \leq 6$), increasing $m$ leads to higher overall accuracy. However, for larger values of $m$ ($m > 6$), further increases in $m$ negatively impact overall accuracy. A similar trend is observed in the many-shot, medium-shot, and few-shot accuracies (see Figures 3a, 3b, and 3c for details). This occurs because a small value of $m$ enhances the model's ability to capture class-level semantics, whereas a large value introduces data imbalance within the feature spaces. Based on these experimental results, the value of $m$, by default, is set to 6 in the real long-tailed scenario.

**Impact of Fixed Sample Size.** We conduct experiments on CIFAR-100-LT to explore the impact of fixed sample size (*i.e.*, $k$) on imbalanced classification. In the experiments, we gradually increase the value of $k$ from 5 to 100. Note that the value of $k$ can be set to larger than the sample size of the minority class. In this case, the actual sample size is set to $\min(k, N_c)$, where $N_c$ is the sample size of the minority class. Figure 4 shows the experimental results. It is observed that a large value of $k$ consistently benefits the many-shot accuracy (see Figure 4a) but hurts the few-shot accuracy (see Figure 4c). This is because increasing the value of $k$ corresponds to adding additional majority samples, but it also enlarges the data imbalance between the majority and minority classes. Figure 4d

Table 10: Performance results on CIFAR-10 under various temperature parameters, with the best results shown in bold

| Temperature Parameter | Many-shot | Medium-shot | Few-shot | Overall |
|:---:|:---:|:---:|:---:|:---:|
| $\tau = 0.1$ | 65.1 | **66.2** | **73.0** | 67.7 |
| $\tau = 0.3$ | 69.4 | 66.0 | 67.6 | **68.2** |
| $\tau = 0.5$ | 72.3 | 64.7 | 57.4 | 66.3 |
| $\tau = 0.7$ | **73.1** | 62.6 | 56.8 | 66.1 |

Table 11: Average sample size after DPP sampling across different ground set sizes

| Ground Set Size ($N$) | 100 | 500 | 1000 | 2000 | 5000 |
|:---:|:---:|:---:|:---:|:---:|:---:|
| Sample Size of Subset | 31.4 | 147.0 | 308.5 | 606.9 | 1548.4 |
| Percentage of $N$ | 31.4% | 29.4% | 30.9% | 30.3% | 31.0% |

demonstrates that the best trade-off is achieved when the value of $k$ is set to $50$, resulting in the highest overall accuracy of $52.4\%$. We emphasize that when $k$ is set to 50, the ratio $\frac{k}{N_C} = 10$, where $N_C = 5$ represents the sample size of the smallest class in CIFAR-100-LT. Thus, in practical scenarios, the value of $k$ is typically set to $10N_C$ by default.

**Influence of Temperature Parameter.** This section investigates the influence of the BNS temperature parameter $\tau$ on representation learning. Specifically, we conduct experiments on CIFAR-10-LT with an imbalance factor of 100, varying temperature parameters from $0.1$ to $0.7$ in steps of $0.2$. The results, summarized in Table 10, are reported as linear probing accuracy.

We observe that smaller values of $\tau$ (*e.g.*, $0.1$) improve performance on tail classes (Few-shot) but reduce accuracy on head classes (Many-shot). Conversely, larger values of $\tau$ increase Many-shot accuracy at the expense of Few-shot performance. This behavior occurs because larger $\tau$ values soften the contrastive loss, giving more weight to hard negative samples—typically dominated by head-class instances—thereby favoring head classes during representation learning. The best trade-off between head and tail performance is achieved when $\tau$ is set to $0.3$, where the accuracy gap between Many-shot and Few-shot categories is minimized (*i.e.*, $1.8\%$).

### B.7 The Expected Subset Sample Size after a DPP

This section supports Appendix A.7 by presenting the empirical results on the expected subset sample size after a Determinantal Point Process (DPP). The Theorem A.5 states that for a ground set containing $N$ samples, its expected number of elements selected by a DPP under a uniform eigenvalue distribution assumption is approximately $N(1 - \ln 2) \approx 30.7\%$. To empirically validate this claim, we ran the DPP over the CIFAR-10-LT dataset with its ground set size ranging from $N = 100$ to $N = 5000$. Table 11 presents the empirical results (averaging over 10 trials). It is observed that the resulting subset sizes consistently align with the theoretical prediction, *i.e.*, $N(1 - \ln 2)$, across all tested values of $N$. These results suggest that the uniform eigenvalue assumption—while idealized—does not significantly distort the practical behavior of a DPP.

## C  Broader Impact

Our proposed methods address the challenging problems regarding the long-tailed data distributions, envisioned to make significant contributions to machine learning (ML) research and advance its applications across various critical domains. *First*, real-world data often exhibit long-tailed or imbalanced distributions, posing significant challenges for conventional ML algorithms. These issues are prevalent in critical applications such as social network spam detection, online transaction fraud detection, medical diagnosis, and others. Misclassifications in these domains can lead to catastrophic consequences, such as undetected fraudulent activities or delayed medical treatments. By effectively addressing the challenge of long-tail data distributions, our method enables state-of-the-art ML techniques to perform reliably in critical domains, fostering improved decision-making and delivering meaningful societal benefits. *Second*, our proposed two-stage learning framework comprises Balanced Negative Sampling (BNS) for representation learning and Information-Preservable Determinantal Point Process (IP-DPP) for rectifying biased classifiers. These components are not only effective on their own but can also be seamlessly integrated with existing state-of-the-art methods. Our

experiments reveal that integrating BNS or IP-DPP into existing methods significantly improves their performance on long-tailed data. This broad adaptability enables a more inclusive application of ML techniques across diverse datasets and tasks, providing a practical tool for practitioners in various industries.

