# OpenReview forum: "Long-Tailed Recognition via Information-Preservable Two-Stage Learning"
_NeurIPS.cc/2025/Conference — NeurIPS 2025 spotlight_

### Official Review · Reviewer_VmnD · 2025-06-23

**Clarity:** 2
**Significance:** 3
**Originality:** 3
**Rating:** 4
**Confidence:** 4

**Summary:**

This paper addresses the challenge of long-tailed recognition by proposing a novel two-stage learning framework that preserves valuable dataset information while mitigating class imbalance. In the first stage, the authors introduce Balanced Negative Sampling (BNS), a representation learning strategy grounded in information theory. BNS is shown to maximize mutual information between augmented views, which is theoretically equivalent to minimizing intra-class distances, thus leading to well-separated feature spaces. In the second stage, the authors propose Information-Preservable Determinantal Point Process (IP-DPP), a novel undersampling technique that prioritizes mathematically informative samples to correct majority-biased decision boundaries without degrading overall model performance. Extensive experiments on four benchmark datasets validate the effectiveness of the proposed method.

**Questions:**

Please see the "Strengths And Weaknesses" part.

**Ethical Concerns:**

["NO or VERY MINOR ethics concerns only"]

**Final Justification:**

Dear authors,

Thank you for your thoughtful response and clarification. I appreciate the explanation regarding how the proposed loss function combines both instance-level and class-level semantics, which makes it more effective than other CL-based methods in long-tailed learning scenarios. Your other responses also satisfactorily address my remaining concerns. All my questions have been resolved, and I will maintain my score of 4.

**Limitations:**

yes.

**Quality:**

3

**Strengths And Weaknesses:**

**Strengths:**

- The authors provide a theoretical analysis of the proposed method, enhancing its soundness and credibility.
- Extensive experiments are conducted, including performance comparisons and ablation studies, to validate the effectiveness of the method, which strengthens its overall reliability.
- The results demonstrate that the proposed method is effective, achieving substantial performance gains.



**Weaknesses:**

- The paper would benefit from more explanation of why the BNS method is particularly effective for learning representations under long-tailed scenarios. It would be helpful to elaborate on the differences and advantages of the proposed implementation compared to standard contrastive learning (CL) methods, as well as other CL-based long-tailed learning methods such as KCL, TSC, and SBCL.
- More representative long-tailed learning methods, such as PaCo$^1$, BCL$^2$, DDC$^3$, GPaCo$^4$ and DirMixE$^5$, should be included in the related work for a more comprehensive comparison.
- There are also some typos, such as in line 161, where "presentation space" should be "representation space".

-----

$^1$ Parametric contrastive learning, ICCV 2021

$^2$ Balanced Contrastive Learning for Long-Tailed Visual Recognition, CVPR 2022

$^3$ A Unified Generalization Analysis of Re-Weighting and Logit-Adjustment for Imbalanced Learning, NeurIPS 2023

$^4$ Generalized Parametric Contrastive Learning, TPAMI 2023

$^5$ Harnessing Hierarchical Label Distribution Variations in Test Agnostic Long-tail Recognition, ICML 2024

---

> ### Author Rebuttal · Authors · 2025-07-29
>
> Thank you for your valuable feedback and for recognizing the contributions of our work.
>
> **Q1. Compared to standard contrastive learning (CL) methods and CL-based long-tailed learning approaches, why BNS is particularly effective in long-tailed data?**
>
> **R1.** BNS is particularly effective because it jointly captures instance-level and class-level semantics, both of which are essential for balanced representation learning in long-tailed settings. To illustrate, we decompose the BNS loss, i.e., Eq. (12) in the main paper, as:
> $$
> \mathcal{L}_{\textrm{BNS}} = - \frac{1}{m+1} \left [
> \underbrace{ \log \sigma (\frac{\mathbf{q}\_{i}^{\top} \mathbf{v}\_{j,i}^{+}}{\tau})
>         + \sum\_{j=1}^{n} \log \sigma ( - \frac{\mathbf{q}\_{i}^{\top} \mathbf{v}\_{j}^{-}}{\tau})}\_{\textrm{instance-level}}  +  \underbrace{ \sum\_{q\_{k} \in \mathbf{Q}\_{i,m}^{+}}
>         \left [ \log \sigma (\frac{\mathbf{q}\_{k}^{\top} \mathbf{v}\_{j,i}^{+}}{\tau})
>         + \sum\_{j=1}^{n} \log \sigma ( - \frac{\mathbf{q}\_{k}^{\top} \mathbf{v}\_{j}^{-}}{\tau}) \right] }\_{\textrm{class-level}}
> \right]
> $$
>
> - The instance-level term ensures fine-grained discrimination between individual samples, similar to standard CL.
> - The class-level term enforces intra-class cohesion, which is critical for tail classes with limited samples.
>
> Standard CL methods such as SimCLR lack explicit class-level alignment, while existing CL-based long-tailed approaches such as KCL, TSC, and SBCL emphasize class-level semantics but often let head-class samples dominate the feature space, weakening instance-level distinction for tail classes. BNS directly optimizes both levels, resulting in more robust and balanced representations that generalize well across both head and tail classes.
>
>
>
> **Q2. Please include more representative long-tailed learning methods, such as PaCo, BCL, DDC, GPaCo, and DirMixE, in the Related Work section.**
>
> **R2.** Thank you for the valuable suggestion. In our revision, we will update the Related Work section to discuss how our method relates to, and differs from, the representative long-tailed learning approaches you mentioned—namely PaCo, BCL, DDC, GPaCo, and DirMixE.
>
>
>
> **Q3. There are also some typos, such as in Line 161, where "presentation space" should be "representation space".**
>
> **R3.** Thank you for pointing this out. We will carefully proofread the manuscript and correct all typos, including the one you identified in Line 161, in the manuscript.

---

> ### Author Response · Authors · 2025-08-04
>
> Dear Reviewer VmnD,
>
> Thank you for taking the time to review our rebuttal and for updating your score accordingly. We truly appreciate your thoughtful evaluation and feedback.
>
> Best regards,
>
> Authors of Paper 5854

---

### Official Review · Reviewer_2ts7 · 2025-06-25

**Clarity:** 2
**Significance:** 3
**Originality:** 4
**Rating:** 4
**Confidence:** 4

**Summary:**

To solve the long-tailed imbalance classification problem, the authors propose a novel two-stage learning approach to mitigate such a majority-biased tendency while preserving valuable information within datasets.

In the first stage, the work aims to learn an effective and well-separated feature space for long-tailed recognition, by maximizing mutual information between instances sharing the same label—a process mathematically equivalent to minimizing intra-class distances. In the second stage, the authors propose Information-Preservable Determinantal Point Process (IP-DPP), aiming to rectify majority-biased classification decision boundaries while maintaining the model’s overall performance.

**Questions:**

What is the relationship between the first and second stages? Is there any interaction between the two stages? What parameters are trained in the second stage? What is the L-ensemble as stated in the second stage?

How it performs if the authors compare the method with the clip-based long-tailed methods? Since CLIP is pretrained based on the contrastive learning, it may provide a powerful backbone.

What type of long-tailed methods can be combined with the proposed method except for the representation-based methods?


As shown in Table2, the few-shot results in iNaturalist 2018 even outperforms the Many-shot and Medium-shot, what's the reason? How to achieve a balance between these groups in this method?

In terms of Table 5, why experimental results on ImageNet-LT and iNaturalist 2018 using various model architectures seem similar?

**Ethical Concerns:**

["NO or VERY MINOR ethics concerns only"]

**Final Justification:**

The authors have addressed my concerns.

**Limitations:**

The second stage is somewhat difficult to understand. It would be better if the authors can provide an overall training workflow and the discussion in the end of method. Please see weakness and questions.

**Paper Formatting Concerns:**

Figure 2 is in front of Figure 1.

**Quality:**

4

**Strengths And Weaknesses:**

Strengths: This work has addressed the challenging long-tailed data classification problem, by proposing a novel information-preservable two-stage learning approach.
Weaknesses: It would be better if the authors can provide a comparison with the clip-based long-tailed methods since the vision encoder is learned in a contrastive way.

It would be better if the authors can discuss the limitations of their proposed work.

---

> ### Author Rebuttal · Authors · 2025-07-29
>
> We greatly appreciate your constructive feedback and your positive evaluation of our work.
>
> **Q1. What is the relationship between the first and second stages? Is there any interaction between the two stages? What parameters are trained in the second stage? Could the authors provide an overall training workflow?**
>
> **R1.** Our framework follows a two-stage training strategy commonly used in long-tailed learning. A deep neural network (DNN) for classification is typically composed of a feature extractor and a linear classifier, and we leverage this architecture as follows:
>
> - Stage 1 (Representation Learning with BNS):
>   We apply our proposed BNS to pre-train the feature extractor on the **entire imbalanced dataset**. This stage focuses on learning generalizable and well-structured feature representations across both head and tail classes.
> - Stage 2 (Classifier Fine-tuning with IP-DPP):
>   In this stage, we fine-tune **both the feature extractor and the linear classifier** on a **class-balanced subset** sampled via our IP-DPP. This step is designed to correct the biased decision boundary introduced by the imbalance in Stage 1.
>
> Although the two stages are not jointly optimized, they are closely related: the success of Stage 2 depends on the quality of the representations learned in Stage 1. Without strong feature representations from BNS, training on a limited balanced subset would not be sufficient to generalize well.
>
> We summarize the full training workflow below:
>
> 1. Input: An imbalanced training dataset.
> 2. Stage 1: Train the feature extractor using BNS on the entire dataset.
> 3. Stage 2: Use IP-DPP to select a balanced subset. Fine-tune the entire network (feature extractor + classifier) on this subset using cross-entropy loss (or long-tailed learning loss, e.g., Focal Loss).
>
>
>
> **Q2. What is the L-ensemble stated in the second stage?**
>
> **R1.** An L-ensemble is a standard formulation of a Determinantal Point Process (DPP), where the probability of selecting a subset of items is defined using a positive semi-definite matrix called the L matrix. This matrix jointly captures the individual quality of items and their pairwise diversity.
>
> In the context of our second-stage sampling strategy, the L-ensemble is constructed based on **information content**, allowing our IP-DPP method to prioritize mathematically informative samples. This enables the selection of balanced and representative subsets that effectively correct biased decision boundaries—without sacrificing critical information—especially in long-tailed settings.
>
>
>
> **Q3. Can the authors compare the proposed method with CLIP-based long-tailed methods?**
>
> **R3.** We conducted additional experiments on iNaturalist 2018, comparing our method with **VL-LTR** [1], a recent CLIP-based approach designed for long-tailed recognition.
>
> Table A presents the comparative results. We observe that VL-LTR achieves higher Many-shot accuracy, likely due to its ability to exploit rich visual-textual pairs available for head classes. In contrast, our method outperforms VL-LTR on Medium-shot, Few-shot, and Overall accuracy, with a particularly significant improvement of +5.6% in Few-shot accuracy, demonstrating the strength of our approach in the long-tailed setting.
>
> These findings suggest that our BNS and IP-DPP methods effectively handle long-tailed distributions without reliance on external textual supervision. Nonetheless, we believe integrating CLIP-based backbones or leveraging textual information could further enhance our method's performance, especially in extreme imbalanced settings. We will explore this promising direction in our future work.
>
>
>
> **Table A: Comparison with CLIP-based long-tailed method on iNaturalist 2018**
>
> | Method | Many-shot | Medium-shot | Few-shot | Overall  |
> | :----: | :-------: | :---------: | :------: | :------: |
> | VL-LTR | **75.3**  |    72.6     |   69.9   |   71.8   |
> |  Ours  |   72.7    |  **72.9**   | **75.5** | **74.0** |
>
>
>
> [1] VL-LTR: Learning Class-wise Visual-Linguistic Representation for Long-Tailed Visual Recognition, ECCV 2022.
>
>
>
> **Q4. What type of long-tailed methods can be combined with the proposed method, except for representation-based methods?**
>
> **R4.** Our proposed methods— BNS and IP-DPP—are highly flexible and can be integrated with a broad range of long-tailed learning techniques beyond representation-based methods.
>
> Specifically:
>
> - BNS can be readily combined with **re-weighting approaches** such as Focal Loss and LDAM Loss, as well as **sampling-based strategies** like SMOTE and Random Undersampling.
> - IP-DPP can be integrated with standard or long-tailed **contrastive learning frameworks** such as SimCLR, KCL, TSC, and SBCL, as well as aforementioned re-weighting methods.
>
> We provide empirical evidence in Table 4 of the Appendix, which reports results on CIFAR-10-LT and CIFAR-100-LT using combinations of BNS and IP-DPP with some of the aforementioned approaches. The consistent performance improvements across these combinations demonstrate the broad compatibility and utility of our two key components.
>
>
>
> **Q5. Why does the Few-shot accuracy on iNaturalist 2018 outperform the Many-shot and Medium-shot accuracies? How can a balanced performance across Many-shot, Medium-shot, and Few-shot categories be achieved?**
>
> **R5.** Thank you for your thoughtful question. The observation arises primarily due to the **high imbalance factor (i.e., IF=500)** present in the iNaturalist 2018 dataset. Our IP-DPP method aims to mitigate this imbalance by sampling a more **balanced subset** in the second stage of training.
>
> In particular, we fix a **uniform sample size $k$** across all classes during IP-DPP sampling. While this strategy helps elevate the performance on **few-shot classes**, it inherently reduces the number of samples from **many-shot** and **medium-shot** classes—leading to a drop in their respective accuracies.
>
> To achieve a more balanced performance across all three groups, we can fine-tune the per-class sample size $k$. By adjusting $k$ (e.g., through class-aware or adaptive sampling strategies), we can trade off between the extremes and improve overall class-wise balance.
>
> We plan to explore more adaptive subset selection strategies in future work to dynamically balance performance across many-shot, medium-shot, and few-shot classes.
>
>
>
> **Q6. In Table 5, the experimental results on ImageNet-LT and iNaturalist 2018 appear similar across different model architectures. Could the authors explain why the performance is relatively consistent despite the architectural differences?**
>
> **R6.** Thank you for the insightful observation. The relatively consistent performance across architectures on ImageNet-LT and iNaturalist 2018 can be attributed to two main factors. First, both datasets contain a large number of similar animal categories, e.g., birds, mammals, and insects, which encourages models to learn comparable visual features across architectures. Second, both datasets exhibit similar long-tailed distributions due to their web-based collection process. Common species tend to dominate the head classes, while rare species form the tail, creating similar imbalance patterns that influence model behavior more than architectural differences.
>
>
>
> **Q7. Please discuss the limitations of the proposed work.**
>
> **R7.** The main limitation of our proposed method lies in the additional computational overhead introduced by our IP-DPP sampling strategy, which involves non-trivial matrix operations (see Tables 6 and 7 in the Appendix for computational overhead analysis). While this overhead is manageable in our experiments, it may become a concern in large-scale, real-world datasets.
>
> In our revision, we will include a dedicated Limitations section to discuss this issue in detail and outline potential directions for improving the efficiency of IP-DPP.
>
>
>
> **Q8. Figure 2 appears before Figure 1 in the paper. Please revise the ordering for consistency.**
>
> **R8.** Thank you for pointing this out. We will fix this in our revision.

---

> > ### Comment · Reviewer_2ts7 · 2025-08-03
> >
> > Thanks for your detailed response to my questions.
> >
> > Reviewer

---

> ### Author Response · Authors · 2025-08-04
>
> Dear Reviewer 2ts7,
>
> Thank you for taking the time to review our rebuttal. We would greatly appreciate it if you could let us know whether you have any remaining questions or comments. If so, we would be happy to address them before the end of the Author-Reviewer Discussion phase (August 6, 11:59 PM AoE).
>
> If our responses have addressed your concerns, we would be grateful if you could consider updating your score accordingly.
>
> Best regards,
>
> Authors of Paper 5854

---

### Official Review · Reviewer_bC3v · 2025-06-29

**Clarity:** 4
**Significance:** 2
**Originality:** 2
**Rating:** 5
**Confidence:** 4

**Summary:**

This paper addresses the challenge of class imbalance in real-world datasets by introducing a novel two-stage learning framework. The first stage employs an information-theoretic representation learning method that effectively minimizes intra-class distances to create well-separated feature spaces. The second stage introduces an innovative undersampling technique that selects informative samples to correct biased decision boundaries caused by majority classes, without sacrificing overall model performance. Extensive experiments on long-tailed benchmark datasets demonstrate that this approach achieves state-of-the-art results.

**Questions:**

1.How do the limitations of traditional representation learning manifest? Why can't the feature learning stage effectively distinguish between head and tail classes?

2.Is it possible to visualize how Balanced Negative Sampling (BNS) improves representation specifically for tail classes?

 If the authors are able to address the above issues, I would be willing to raise my score.

**Ethical Concerns:**

["NO or VERY MINOR ethics concerns only"]

**Final Justification:**

The author's answers to my questions are very clear, including the ablation experiments on temperature and other supplementary experiments. At the same time, the author also resolved my confusion in the theoretical part.I believe these methods are very important for evaluating the completeness of this paper, so I have decided to raise my score.

**Limitations:**

yes

**Quality:**

3

**Strengths And Weaknesses:**

Strengths :
The author's reasoning is very clear and easy to understand, with a well-defined motivation. The overall flow is smooth, and the ablation studies are comprehensive.

Weaknesses:

Tables 1 and 2 do not show results under different class balance ratios. The appendix only reports overall performance on CIFAR datasets. It would be beneficial to present more detailed results showing the trade-off between head and tail class performance at varying balance ratios.

The effect of the initial value of the temperature parameter τ on results is not investigated (e.g., how changes in τ impact accuracy).

---

> ### Author Rebuttal · Authors · 2025-07-29
>
> We sincerely appreciate your thoughtful feedback and recognition of our work.
>
> **Q1. How do the limitations of traditional representation learning manifest? Why can't the feature learning stage effectively distinguish between head and tail classes?**
>
> **R1.** Effective representation learning for long-tailed data must capture two key aspects: (i) **instance-level semantics** for high-quality feature representations; and (ii) **class-level semantics** for well-separated feature spaces. Traditional contrastive learning methods such as SimCLR and MoCo focus solely on instance-level discrimination, treating each sample as its own class. While this works well under balanced distributions, it lacks mechanisms for class-level alignment, which is crucial in long-tailed scenarios. As a result, head-class features dominate the space, while tail-class instances—due to data scarcity—fail to form coherent clusters, resulting in poor generalization.
>
> Moreover, these methods lack effective sampling strategies and often fail to retrieve sufficient positive pairs for tail classes, further degrading their representation quality and pushing them to ambiguous regions of the feature space.
>
> Our proposed BNS method addresses these limitations by jointly modeling both instance-level and class-level semantics. The BNS loss, i.e., Eq. (12) in the paper, can be decomposed as:
> $$
> \mathcal{L}_{\textrm{BNS}} = - \frac{1}{m+1} \left [\underbrace{ \log \sigma (\frac{\mathbf{q}\_{i}^{\top} \mathbf{v}\_{j,i}^{+}}{\tau})         + \sum\_{j=1}^{n} \log \sigma ( - \frac{\mathbf{q}\_{i}^{\top} \mathbf{v}\_{j}^{-}}{\tau})}\_{\textrm{instance-level}}  +  \underbrace{ \sum\_{q\_{k} \in \mathbf{Q}\_{i,m}^{+}}         \left [ \log \sigma (\frac{\mathbf{q}\_{k}^{\top} \mathbf{v}\_{j,i}^{+}}{\tau})        + \sum\_{j=1}^{n} \log \sigma ( - \frac{\mathbf{q}\_{k}^{\top} \mathbf{v}\_{j}^{-}}{\tau}) \right] }\_{\textrm{class-level}}\right]
> $$
>
> - The **instance-level** term maintains fine-grained distinctions between individual samples.
>
> - The **class-level** term promotes intra-class cohesion, especially important for tail classes.
>
> Furthermore, BNS constructs a balanced neighborhood by sampling an additional $m$ positive pairs from the same class, ensuring sufficient positive context even for underrepresented classes. This dual-level alignment results in a more structured, balanced feature space that improves generalization across both head and tail classes.
>
>
>
> **Q2. Is it possible to visualize how Balanced Negative Sampling (BNS) improves representation specifically for tail classes?**
>
> **R2.** Thank you for your question. A t-SNE visualization focused solely on tail classes is an effective way to demonstrate this improvement.
>
> While the NeurIPS rebuttal policy restricts us from adding new figures via external links, we would like to highlight that Figures 2(a) and 2(c) in our submission already provide relevant insights.
>
> Specifically, Figure 2(c) shows the feature space learned using BNS. Notably, the top-most (gold) and left-most (cyan) clusters correspond to two tail classes. These clusters exhibit strong intra-class compactness and are well-separated from nearby head classes (depicted in orange and blue), indicating that BNS facilitates more discriminative and robust representations for tail classes.
>
> In contrast, Figure 2(a), which shows representations from SBCL, displays substantial overlap between head and tail classes. The head classes dominate the feature space, leading to poorer separability for tail classes. This qualitative comparison demonstrates that BNS significantly enhances the representation quality for tail classes by promoting both intra-class compactness and inter-class separability.
>
>
>
> **Q3. It would be beneficial to present more detailed results showing the trade-off between head and tail class performance at varying balance ratios.**
>
> **R3.** Thank you for the valuable suggestion. To provide a more comprehensive evaluation of our method under different imbalance scenarios, we report detailed results on CIFAR-10-LT and CIFAR-100-LT, broken down into Many-shot, Medium-shot, Few-shot, and Overall accuracy. These results are summarized in Tables A and B below.
>
> Our findings show that as the imbalance factor increases, performance on Many-shot (head) classes remains relatively stable, whereas accuracy on Few-shot (tail) classes declines significantly. This trend reflects the growing representation gap between head and tail classes under more severe imbalance, which makes learning effective representations for tail classes increasingly difficult. These extended results further highlight the importance of addressing data imbalance and will be included in the final manuscript.
>
>
> **Table A: Performance on CIFAR-10-LT under varying imbalance factors**
>
> | Imbalanced Factor (IF) | Many-shot | Medium-shot | Few-shot | Overall |
> | :--------------------: | :-------: | :---------: | :------: | :-----: |
> |         IF=50          |   85.3    |    81.4     |   72.2   |  81.9   |
> |         IF=100         |   82.0    |    76.3     |   67.2   |  76.4   |
> |         IF=200         |   81.4    |    75.9     |   64.4   |  73.5   |
>
>
>
> **Table B: Performance on CIFAR-100-LT under varying imbalance factors**
>
> | Imbalanced Factor (IF) | Many-shot | Medium-shot | Few-shot | Overall |
> | :--------------------: | :-------: | :---------: | :------: | :-----: |
> |         IF=50          |   63.7    |    57.4     |   34.7   |  55.9   |
> |         IF-100         |   62.4    |    59.7     |   31.9   |  52.4   |
> |         IF=200         |   62.8    |    58.9     |   24.2   |  46.7   |
>
>
>
>
>
> **Q4. Please include ablation studies on the temperature parameter $\tau$.**
>
> **R4.** We conducted an ablation study on the temperature parameter $\tau$, with results presented in Table C below. The study was performed on CIFAR-10-LT with an imbalance factor of 100, varying $\tau$ from 0.1 to 0.7 in steps of 0.2.
>
> We observe that smaller values of $\tau$ (e.g., 0.1) improve performance on tail classes (Few-shot) but reduce accuracy on head classes (Many-shot). Conversely, larger values of $\tau$ increase Many-shot accuracy at the expense of Few-shot performance. This behavior occurs because larger $\tau$ values soften the contrastive loss, giving more weight to hard negative samples—typically dominated by head-class instances—thereby favoring head classes during representation learning.
>
> The best trade-off between head and tail performance is achieved when $\tau$ is set to 0.3, where the accuracy gap between Many-shot and Few-shot categories is minimized. We will include these findings in our manuscript.
>
>
>
> **Table C: Ablation study on the temperature parameter $\tau$ (CIFAR-10-LT, IF = 100)**
>
> | Temperature Parameter | Many-shot | Medium-shot | Few-shot | Overall  |
> | :-------------------: | :-------: | :---------: | :------: | :------: |
> |     $\tau$ = 0.1      |   65.1    |  **66.2**   | **73.0** |   67.7   |
> |     $\tau$ = 0.3      |   69.4    |    66.0     |   67.6   | **68.2** |
> |     $\tau$ = 0.5      |   72.3    |    64.7     |   57.4   |   66.3   |
> |     $\tau$ = 0.7      | **73.1**  |    62.6     |   56.8   |   66.1   |

---

> ### Author Response · Authors · 2025-08-04
>
> Dear Reviewer bC3v,
>
> Thank you again for taking the time to read our rebuttal. We would greatly appreciate it if you could let us know whether our responses have adequately addressed your questions and concerns—particularly those raised in Q1 and Q2 in the “Questions” section.
>
> If our clarifications have resolved your concerns, we would be grateful if you would consider raising your score accordingly. If there are any remaining issues or questions, please do not hesitate to share them—we would be happy to address them before the end of the Author-Reviewer Discussion phase (August 6, 11:59 PM AoE).
>
> Best regards,
>
> Authors of Paper 5854

---

> > ### Comment · Reviewer_bC3v · 2025-08-04
> >
> > Yes, your response has fully dispelled my concerns. I have made changes to my score, and thank you again.

---

> > > ### Author Response · Authors · 2025-08-04
> > >
> > > Thank you for your follow-up and for updating your score—we truly appreciate it.

---

### Official Review · Reviewer_8YK9 · 2025-07-03

**Clarity:** 3
**Significance:** 3
**Originality:** 3
**Rating:** 5
**Confidence:** 4

**Summary:**

This paper proposes a novel two-stage learning framework to address the challenges of long-tailed data distributions in classification tasks. The authors identify two major issues in long-tailed settings: representation learning bias and classifier bias. To tackle these, they introduce:

Balanced Negative Sampling (BNS): A contrastive representation learning strategy that constructs semantically rich and balanced feature spaces by carefully selecting informative negative samples. BNS captures both instance- and class-level semantics, enabling better feature separation, especially for medium- and few-shot classes.

Information-Preservable Determinantal Point Process (IP-DPP): A sampling method used during classifier training to select a diverse and representative subset of the training data. IP-DPP maintains a balance between classes while preserving informative instances and rectifying biased decision boundaries.

Extensive experiments are conducted on CIFAR-10-LT, CIFAR-100-LT, ImageNet-LT, and iNaturalist 2018 datasets. The proposed method achieves state-of-the-art performance and shows robustness across different imbalance levels and model architectures. The paper also includes theoretical insights, ablation studies, computational cost analysis, and discussions on hyperparameter sensitivity. The framework’s modularity and compatibility with existing methods make it a promising solution for real-world imbalanced learning scenarios.

**Questions:**

1. Clarification on the Statistical Assumption in DPP Derivation

In Theorem A.5, the expected sample size of a DPP is derived under the assumption that eigenvalues follow a uniform distribution in [0, 1]. This is a strong assumption and may not reflect the actual eigenvalue distribution in practice. Could the authors provide empirical evidence (e.g., histogram of eigenvalues) to support the validity of this assumption on real datasets? Alternatively, could the authors comment on how sensitive the performance is to this assumption?

2. Ablation on the Interaction Between BNS and IP-DPP

The paper presents ablations for BNS and IP-DPP independently. However, it is unclear how these two components interact. Could the authors provide an ablation showing the incremental benefit of BNS and IP-DPP when added sequentially (e.g., base → base+BNS → base+BNS+IP-DPP)? This would help isolate their individual and combined contributions more clearly.

**Ethical Concerns:**

["NO or VERY MINOR ethics concerns only"]

**Limitations:**

Yes

**Quality:**

4

**Strengths And Weaknesses:**

Strengths:

This paper proposes a novel two-stage learning framework to address long-tailed recognition, integrating Balanced Negative Sampling (BNS) for improved representation learning and Information-Preservable Determinantal Point Process (IP-DPP) for balanced classifier training. The approach is both theoretically grounded and empirically validated. The authors provide rigorous mathematical derivations, detailed ablation studies, and comprehensive evaluations on multiple standard benchmarks, including CIFAR-10-LT, CIFAR-100-LT, ImageNet-LT, and iNaturalist 2018. Notably, the method consistently outperforms strong baselines and can be easily combined with existing techniques, demonstrating broad applicability. The paper is well-written and clearly structured, with useful visualizations and thorough supplementary material that supports the main contributions. The use of IP-DPP for informative and class-balanced subsampling is particularly innovative, and the combination of DPP theory with long-tailed learning is a valuable contribution.

Weaknesses:

While the theoretical contributions are strong, some derivations (e.g., those involving eigenvalue distributions in DPP) rely on simplifying assumptions (e.g., uniform distribution) that may not always hold in practice, limiting their generality. Some notation in the theoretical sections is dense, and parts of the methodology (e.g., details of the BNS selection mechanism) could benefit from more intuitive explanation.

Overall Assessment:

This is a high-quality, well-motivated, and thorough piece of work that offers solid improvements and insights into long-tailed recognition problems.

---

> ### Author Rebuttal · Authors · 2025-07-29
>
> We are grateful for your constructive feedback and your appreciation of our work.
>
> **Q1. Theorem A.5 assumes a uniform distribution of eigenvalues. Could the authors provide empirical evidence (e.g., histogram of eigenvalues) to support the validity of this assumption on real datasets? Or could the authors comment on how sensitive the performance is to this assumption?**
>
> **R1.** Thank you for your question. We appreciate your close examination of our theoretical assumptions.
>
> Due to the NeurIPS rebuttal policy, which prohibits the inclusion of new figures via anonymous links, we are unable to provide a histogram of eigenvalues at this time. However, we provide more direct empirical evidence to support the practical validity of the uniform eigenvalue assumption used in Theorem A.5.
>
> Theorem A.5 states that for a ground set of size $N$, the expected number of elements selected by a DPP (Determinantal Point Process) under a uniform eigenvalue distribution assumption is approximately $N(1 - \ln 2) \approx \mathbf{30.7 \\%}$. To empirically validate this claim, we ran DPP sampling over the CIFAR-10-LT dataset with ground set sizes ranging from $N=100$ to $N=5000$, averaging the results over 10 trials per setting.
>
> As shown in Table A below, the resulting subset sizes consistently align with the theoretical prediction across all tested values of $N$, suggesting that the uniform eigenvalue assumption—while idealized—does not significantly distort the practical behavior of the DPP. We will include additional empirical analyses, including eigenvalue histograms on real datasets, in the camera-ready version to further substantiate this assumption.
>
>
>
> **Table A. Empirical average sample size after DPP sampling across different ground set sizes**
>
> | Ground Set Size ($N$) |  100  |  500  | 1000  | 2000  |  5000  |
> | :-------------------: | :---: | :---: | :---: | :---: | :----: |
> | Sample Size of Subset | 31.4  | 147.0 | 308.5 | 606.9 | 1548.4 |
> |   Percentage of $N$   | 31.4% | 29.4% | 30.9% | 30.3% | 31.0%  |
>
>
>
>
> **Q2. Could the authors provide an ablation showing the incremental benefit of BNS and IP-DPP?**
>
> **R2.** Thank you for your comments. We separated BNS and IP-DPP in the original ablation study because they serve different purposes and are evaluated under different metrics. Specifically, BNS is designed for representation learning and is evaluated via linear probing accuracy, whereas IP-DPP is a sampling strategy applied during the classifier training stage and evaluated using classification accuracy.
>
> To directly address your question, we conducted an additional ablation study on CIFAR-100-LT with an imbalance factor (IF) of 100. All methods follow a standard two-stage learning strategy:
>
> - **Stage 1**: Representation learning
> - **Stage 2**: Classifier training to mitigate decision boundary bias
>
> For the baseline method, we use SimCLR for Stage 1 and Random Undersampling for Stage 2. We then incrementally improve this setup by: (i) replacing SimCLR with BNS (Stage 1), and  (ii) replacing Random Undersampling with IP-DPP (Stage 2). Finally, we report the performance of our full approach, which combines both BNS and IP-DPP.
>
> The results are shown in Table B below. We make four key observations:
>
> First, **Baseline (SimCLR + Random)** performs worst overall because SimCLR fails to learn effective representations for tail classes, while Random Undersampling discards valuable information during re-balancing.
>
> Second, **BNS + Random** improves Few-shot and Medium-shot performance by learning more structured representations, but still suffers in Many-shot accuracy due to suboptimal classifier training.
>
> Third, **SimCLR + IP-DPP** improves Many-shot performance by preserving informative head-class samples, but continues to struggle with tail classes due to SimCLR’s poor representations.
>
> Fourth, **our method (BNS + IP-DPP)** achieves the best overall performance by combining strong, balanced representations from BNS and informative sample selection from IP-DPP to reduce bias during classifier training.
>
>
>
> **Table B: Ablation studies on the incremental benefit of BNS and DP-DPP (CIFAR-100-LT, IF = 100)**
>
> |          Methods           | Many-shot | Medium-shot | Few-shot | Overall |
> | :------------------------: | :-------: | :---------: | :------: | :-----: |
> | Baseline (SimCLR + Random) |   58.1    |    56.6     |   28.1   |  48.6   |
> |        BNS + Random        |   57.8    |    59.6     |   31.4   |  50.5   |
> |      SimCLR + IP-DPP       |   62.7    |    56.9     |   28.5   |  50.4   |
> |    Ours (BNS + IP-DPP)     |   62.4    |    59.7     |   31.9   |  52.4   |

---

> ### Author Response · Authors · 2025-08-04
>
> Dear Reviewer 8YK9,
>
> Thank you again for your constructive feedback and appreciation of our work. As we approach the end of the Author-Reviewer Discussion phase (August 6, 11:59 PM AoE), we kindly ask if you could take a moment to review our rebuttal.
>
> If you have any remaining questions or comments, we would be happy to address them before the deadline.
>
> Best regards,
>
> Authors of Paper 5854

---

> > ### Comment · Reviewer_8YK9 · 2025-08-04
> >
> > I have read the author's response and the comments from the other reviewers. The author has addressed my questions and concerns.

---

### Comment · Area_Chair_KUTk · 2025-08-02

Dear 5854 Reviewers:  The authors have provided detailed rebuttals to your reviews.  I'd urge you to read their rebuttals early to allow further interactions that help clarify any lingering confusion or misunderstanding.  Thank you!  AC

---

### Note · Authors · 2025-08-12

Dear Reviewers and Area Chair,

We sincerely thank you for your constructive feedback and active engagement throughout the author–reviewer discussion. Based on the exchange, we believe our rebuttal has addressed all questions and concerns, with no unresolved issues remaining.

In our final revision, we will incorporate the additional experiments and analyses conducted during the rebuttal phase to further strengthen the manuscript, including but not limited to:

- **Empirical validation** of the assumption in Theorem A.5 (R1 to Reviewer 8YK9).
- **Ablation studies** on: (i) the incremental contributions of our BNS and IP-DPP methods (R2 to Reviewer 8YK9); and (ii) the effect of the temperature parameter (R4 to Reviewer bC3v).
- **Baseline comparison** with a CLIP-based long-tailed method (R3 to Reviewer 2ts7).

We will also expand the manuscript with the following revisions:

- A detailed discussion of the advantages of our BNS approach over standard contrastive learning and existing long-tailed learning methods (R1 to Reviewer bC3v; R1 to Reviewer VmnD).
- A **Limitations** section outlining current constraints and future research directions (R7 to Reviewer 2ts7).
- Inclusion of representative long-tailed learning methods in the Related Work section (R2 to Reviewer VmnD).

We greatly appreciate the reviewers’ thoughtful feedback, which has helped us enhance both the technical depth and clarity of our work.

Best regards,

Authors of Paper 5854

---

### Decision · Program_Chairs · 2025-09-17

**Decision:**

Accept (spotlight)

**Comment:**

This paper proposes a two-stage method to address class imbalance by learning compact representations and refining decision boundaries through informative undersampling. The approach achieves state-of-the-art performance on long-tailed benchmarks.

The reviewers praised the paper for its well-motivated and clearly presented two-stage framework that tackles long-tailed classification using Balanced Negative Sampling (BNS) and a novel Information-Preservable Determinantal Point Process (DPP) approach. They highlighted the strong theoretical grounding, smooth reasoning, and comprehensive ablation studies. The method consistently outperforms strong baselines across several benchmark datasets and is modular enough to integrate with existing techniques. The use of DPPs for informative, class-balanced subsampling was seen as a particularly creative and valuable contribution.

The reviewers pointed out several areas for improvement. Some derivations, especially those involving eigenvalue distributions in DPPs rely on simplifying assumptions like uniformity that may not hold universally, reducing generality.  Notation can be dense, and the BNS mechanism would benefit from more intuitive explanation. Empirically, results are only shown for fixed class imbalance ratios, with limited analysis of head–tail trade-offs or the influence of key hyperparameters like the temperature τ. Reviewers also requested more clarity on why BNS is effective under long-tailed conditions and how it compares to other contrastive learning methods such as KCL, TSC, and SBCL. Additionally, several strong long-tailed baselines (PaCo, BCL, DDC, GPaCo, DirMixE) were missing from related work. Minor presentation issues like typos and a lack of discussion around limitations were also noted.

The paper presents a strong and well-rounded contribution to long-tailed classification, combining theoretical innovation with solid empirical results. All reviewer concerns were addressed during the rebuttal phase, including additional ablations, empirical validations, and new baseline comparisons (e.g., with CLIP-based methods). The authors have also committed to incorporating these updates, such as expanded analysis of their BNS approach, sensitivity to temperature settings, and a clear discussion of limitations, into the final manuscript. With these revisions, the work not only maintains its technical rigor but also improves in clarity, completeness, and relevance, warranting acceptance.